



# A multi-model analysis of the decadal prediction skill for the North Atlantic ocean heat content

Teresa Carmo-Costa[1], Roberto Bilbao[2], Jon Robson[3], Ana Teles-Machado[1,4], and Pablo Ortega[2]

[1]Faculdade de Ciências da Universidade de Lisboa - Instituto Dom Luiz, Lisbon, Portugal
[2]Barcelona Supercomputing Center, Barcelona, Spain
[3]National Centre for Atmospheric Science, University of Reading, Reading, UK
[4]Instituto Português do Mar e da Atmosfera, Lisbon, Portugal

**Correspondence:** Teresa Carmo-Costa (tmcosta@fc.ul.pt) and Pablo Ortega (pablo.ortega@bsc.es)

**Abstract.** Decadal predictions can skillfully forecast the upper ocean temperature in many regions of world. The North Atlantic in particular shows promising results when it comes to high predictive skill of Ocean Heat Content (OHC). Nevertheless, important regional differences exist across Decadal Prediction Systems, which are explored in this multi-model analysis. Differences are also found in their respective uninitialized historical ensembles, which points to large uncertainties in the externally forced

signals. We analyze eight CMIP6 climate models with comparable ensembles of decadal predictions and historical simulations to document their differences in upper OHC skill, and to investigate if intrinsic model characteristics, such as key mean state biases in the local forcing from the atmosphere or the local stratification, can influence the relative predictive role of external forcings and internal variability. Particular attention has been given to the Labrador Sea and its surroundings, since this is found to be a region where upper OHC has low observational uncertainties, yet high inter-model spread in the upper OHC prediction

tion skill of decadal predictions and historical experiments. Benchmarking mean state properties of the local surface fluxes and stratification against observations, both strongly linked with the simulated upper OHC skill for the historical ensembles, suggests that their multi-model mean provides the most realistic estimate of the true forced signal.

## 1 Introduction

In recent decades, the upper waters of the North Atlantic (NA) Ocean have experienced a general long term warming trend, as consistently revealed by various observational datasets (Wang et al., 2018; Häkkinen et al., 2015; Zanna et al., 2019), with a spatially heterogeneous distribution. The NA warming trend is particularly prominent for a vertically integrated variable such as the ocean heat content (OHC), although it shows considerable uncertainty across observationally-based products in its geographical distribution (Palmer et al., 2017; Häkkinen et al., 2015).

One important regional difference in OHC evolution within the NA can be found in the center of the Subpolar North Atlantic (SPNA), which has been subject to a long-term cooling trend (often labeled as the NA Warming Hole (Rahmstorf et al., 2015;



Drijfhout et al., 2012; Keil et al., 2020)), alongside its large multi-decadal variability (Yeager et al., 2016; Hermanson et al., 2014; Carmo-Costa et al., 2021). The main mechanism proposed to explain the cooling involves a reduction in the northward heat advection, in turn responding to either a shift in the North Atlantic gyre circulation (Piecuch et al., 2017) or a weakening

of the Atlantic Meridional Overturning Circulation (AMOC) (Drijfhout et al., 2012; Robson et al., 2016), which might be of forced origin (Caesar et al., 2021), or simply reflect internal variability at multidecadal timescales (Kilbourne et al., 2022).

Internal variability modes in the North Atlantic region could also explain some of the regional differences in the recent OHC variability. The North Atlantic Oscillation (NAO) is an important driver of AMOC variability through its influence on Labrador Sea deep water formation. Positive NAO phases enhance winter surface cooling and can thus help overcome the local vertical

density stratification, promoting the occurrence of deep ocean mixing events. In addition, interannual NAO variations are also known to force local OHC anomalies across the North Atlantic subpolar gyre mediated via changes in the surface buoyancy fluxes and wind stress (Oldenburg et al., 2021). An illustrative example was the record-breaking cold anomaly that the central SPNA experienced in spring 2015 (commonly referred as the Cold Blob (Yeager et al., 2016; Duchez et al., 2016; Josey et al., 2018)), emerging in response to exceptionally rare (both in terms of magnitude and persistence) positive NAO conditions

(Yeager, 2020; Maroon et al., 2021) which enhanced local heat loss. This extremely cold central SPNA state has been linked to the occurrence of a major heatwave in Central Europe in the summer of 2015 (Duchez et al., 2016; Mecking et al., 2019). More generally, multidecadal variations in North Atlantic upper ocean temperatures have been linked to several climate impacts, from the intensity and frequency of Atlantic hurricane activity and Sahel rainfall (Balaguru et al., 2018; Buckley and Marshall, 2016; Zhang and Delworth, 2006), to hydroclimate and temperature conditions in North America and Europe (Enfield et al.,

2001; Sutton and Hodson, 2005; Kwon et al., 2020; Josey et al., 2018).

Understanding the causes of the regional OHC changes in the North Atlantic, by disentangling the contributions from internal and externally forced variability and the underlying uncertainties is, thus, very important to better anticipate how the climate will evolve in the coming years.

There are two major types of simulations within the Climate Model Intercomparison Project phase 6 initiative (CMIP6

Eyring et al., 2016) that can be jointly used to understand the contributions of external forcings and internal variability processes to the recent climate evolution and to understand the sources of predictability of the North Atlantic's ocean-atmosphere system (Meehl et al., 2014) - namely historical simulations and retrospective decadal predictions. While historical experiments are flagship tier-1 simulations to investigate how the Earth system responds to the historical forcings and evaluate the model performance against observations (Eyring et al., 2016), retrospective decadal predictions are performed under the umbrella of

the Decadal Climate Prediction Project (DCPP; Boer et al., 2016) to assess our ability to skilfully predict climate variations from one year up to a decade ahead. Both historical and DCPP experiments use the same prescribed external radiative forcings, but differ in one major aspect: the decadal predictions are initialized to align their starting conditions with a past observed state, which allows them to benefit, in theory, from the predictability arising from internal variability sources. In contrast, historical ensembles are designed to encapsulate a variety of internal variability states, whose climate effects largely cancel out when considering the ensemble mean to capture the externally forced signal, provided the ensemble is large enough (Milinski et al.,

2020).





In the North Atlantic, there are both qualitative and quantitative differences across decadal prediction systems, when it comes to predictability time range, areas with significant predictive capacity, and magnitude of the associated skill. This is very evident when examining Sea Surface Temperature (SST), for which models tend to rapidly lose skill after the first forecast

60    year (e.g. Langehaug et al., 2022). Likewise, in less predictable areas (e.g., the inflow region from the NA to the Norwegian Sea), most models do not even show any clear benefit of initialization, regardless of forecast range. Multiple efforts have been made to skillfully predict NA's variability of both SST and OHC (Bilbao et al., 2021; Robson et al., 2012; Yeager et al., 2012; Mignot et al., 2016; Keenlyside et al., 2008; Kröger et al., 2018; Pohlmann et al., 2009; Borchert et al., 2018; Robson et al., 2018; Polkova et al., 2023), as well as other related variables like the NAO (Athanasiadis et al., 2020; Smith et al., 2020) or

the Atlantic Multidecadal Variability (AMV) (Borchert et al., 2018; Volpi et al., 2017). These efforts, however, usually focus on individual decadal prediction systems, which might lead to model-dependent results.

During the CMIP5 initiative, there were already some studies (Doblas-Reyes et al., 2013; García-Serrano et al., 2015) that circumvented that issue by using a multi-model ensemble mean to extract the common signal and in this way enhance the predictive skill. Following this approach, Doblas-Reyes et al. (2013) concluded that AMV predictability arises mostly

from internal predictability sources. García-Serrano et al. (2015) went even further and showed that those internal sources (leveraged through initialization) help to better simulate the characteristic AMV horseshoe pattern, as well as the teleconnection mechanism with the West African monsoon. CMIP6 multi-model mean analyses have further confirmed the beneficial impact of initialization for delivering skillful AMV forecasts (e.g., Delgado-Torres et al. (2022)), and also higher skill in the Subpolar North Atlantic region than in the CMIP5 experiments (Borchert et al., 2021), although with a more prominent predictive

role of the external forcings. CMIP6 experiments have also shown that while the multi-model mean outperforms globally the median of the ensemble in terms of predictive, some individual models exhibit higher skill, especially in the North Atlantic (Delgado-Torres et al., 2022).

Understanding the differences in skill across models, and in particular the factors that control and enhance the regional predictability, is essential to inform and improve the next generation of decadal prediction systems. Several multi-model stud-

ies have indeed shown how different model biases in the Labrador Sea, from near surface densities Menary and Hermanson (2018), the upper ocean mean stratification (Ortega et al., 2021; Kim et al., 2023) or the local covariability between temperature and salinity (Menary et al., 2015), can degrade important aspects of the North Atlantic decadal variability, including its predictability.

Aside from the realism of the underlying climate models, predictive skill can also be sensitive to the initialization approach.

Model adjustments and shocks resulting from unbalanced initial conditions have been shown to degrade the forecast skill (Mulholland et al., 2015; Polkova et al., 2023). To mitigate the effects of model drift that develops as the model converges from its initial state (which was constrained by observations) to its own climate attractor, the use of anomaly initialization strategies has become more popular (e.g., Volpi et al., 2017; Müller et al., 2018; Bethke et al., 2021). However, the few studies that have explored the potential benefits of this approach over the standard full-field initialization (Smith et al., 2013; Hazeleger et al.,

2013; Volpi et al., 2017; Kröger et al., 2018), failed to identify consistent improvements throughout the globe.



The main aim of this study is to evaluate the predictive skill of the upper 700m OHC in the North Atlantic in a multi-model context, exploring some of the processes and methodological aspects behind the inter-model differences. In particular, the following questions will be addressed:

– Are there areas in the North Atlantic where the OHC is more predictable? Which of those regions are consistently identified across models? And which regions show larger differences in skill across models?

– How much of the upper OHC skill is attributable to external forcing variations? What are the common features and main differences across models?

– How does the representation of the different local drivers and preconditioners of decadal North Atlantic variability influence our understanding of the predictability sources (internal vs. forced)?

– Can some skill limitations or improvements be associated with specific methodological aspects of the forecast systems (i.e., initialization approach, resolution, etc.)?

This paper is organised as follows: Section 2 describes the observational products, models and simulations used, the criteria for the final ensemble selection, as well as some data processing considerations; Sections 3.1, 3.2, 3.3 and 3.4 present the results in four separate scientific blocks: (i) a multi-model evaluation of the upper OHC skill in the NA Ocean to assess the consistency of the results across models and identify outlier behaviours and regions of interest, (ii) a deeper investigation of the role of external forcings and long-term trends in the upper OHC skill in the Labrador Sea (identified in $i$ as a region of interest for its large inter-model differences); (iii) an inter-model comparison of key Labrador Sea mean state model properties that could potentially condition the local OHC variability and skill; and (iv) an analysis of how those model properties affect the inferred predictive role of external forcings on Labrador Sea OHC variability, introducing some observational references to constrain the inter-model uncertainty. The final Section summarises the main results and discusses them in light of previous studies.

## 2 Data and methods

This analysis considers both historical (named HIST hereafter) and DCPP component A (Boer et al., 2016) retrospective decadal prediction ensembles of the CMIP6 initiative to explore the effects of external forcings and internally generated variability on the observed OHC variability and the ability of current climate models to predict it. We will focus on the OHC in the upper 700m (referred to as OHC700 hereafter) and the main pre-conditioners and large-scale drivers of its regional variability and predictability, paying special interest to the major inter-model differences.

### 2.1 Climate model selection

The model selection was based on three criteria: (1) both the HIST and DCPP experiments were available via the Earth System Grid Federation (ESGF) portal for each model; (2) both the HIST and DCPP experiments were driven with the CMIP6 external



forcings to ensure complete consistency in the forced signals; and (3) in both sets of experiments the relevant output variables for our analyses, such as 3D salinity (*so* in CMIP convention), 3D potential ocean temperature (*thetao*), 2D sea level pressure (*psl*), downward surface heat fluxes (*hfds*) and sea ice concentration (*siconc*), were available at monthly frequency for the period 1960-2014 (which is their overlap period). Also, except for two justified exceptions (details below), models with fewer

than 10 ensemble members for HIST and/or DCPP were excluded. A total of 8 AOGCMs fulfilled all the selection criteria. The final multi-model ensemble is composed of the models and experiments listed in Table 1.

For models that provided more than 10 ensemble members, only the first 10 members were retained to maximize the consistency across models, with the exception of EC-Earth3 for which we used 10 historical simulations performed at the BSC. This ensemble size is the minimum that is recommended for the DCPP-A protocol in (Boer et al., 2016) and is the most com-

130 mon ensemble size across the models considered. Two models contributed with fewer than 10 members to the experiments: HadGEM3-GC31-MM (which only has 4 HIST members) and MPI-ESM1-2-HR (which only has 5 DCPP members with the required model outputs), both with comparatively higher horizontal resolution. They thus contribute to this analysis with 4 and 5 members, respectively. This exception allowed us to assess if there is any added value, either in process representation or predictive skill, in increasing the horizontal resolution. More details on the models considered in this analysis and their

characteristics can be read in Table 1.

## 2.2 Observational references

We used the EN4 version 2.2 ocean temperature and salinity observational dataset (Good et al., 2013) to evaluate the predicted OHC700 and the ocean stratification. Three ocean reanalyses - ECDA3.1 (Chang et al., 2013); ORAS4 (Balmaseda et al., 2013); ORAS5 (Zuo et al., 2019); the same used in Carmo-Costa et al. (2021) - were additionally considered and compared

with EN4 to identify the regions with high and low OHC700 observational uncertainty.

To understand the processes driving local OHC700 skill we analysed additional variables, such as sea-level pressure, surface heat fluxes and sea-ice concentration. To determine how realistically the systems simulate these variables, we compared them with other observationally-based datasets. For the atmospheric variables we used the global atmospheric reanalysis ERA5 (Hersbach et al., 2020), as it provides a complete and physically coherent description of recent atmospheric variability that is

145 constrained by observations. These include monthly sea-level pressure fields (necessary to compute the NAO) and net surface heat fluxes (derived from thermal radiation, surface solar radiation, surface sensible heat flux and surface latent heat flux). Finally, to evaluate the sea-ice concentration we used the monthly fields of HadISST.2.2.0.0 (Titchner and Rayner, 2014, hereinafter simply HadISST).

## 2.3 Data preprocessing

Data from both models and observational products were regridded to a common regular 1ºx1º resolution grid. All model outputs were regridded using the Earth System Model Evaluation Tool (ESMValTool; Righi et al., 2020) versions 2.4.0 to 2.7.0, which was particularly useful for its ability to process all models, experiments, start dates and variables in a consistent way. For other pre-processing tasks that were less computationally intensive, such as the calculation of yearly averages or the regridding of the





ERA5 reference data, the Climate Data Operators tool version 1.9.10 (https://mpimet.mpg.de/cdo) was preferred. Additionally,
we used ESMValTool to compute the OHC700 and potential density anomaly ($\sigma$ - sigma; computed for the reference level of
1000m). The post-processed outputs were then analysed with both the s2dverification/s2dv package (Manubens et al., 2018;
Guemas et al., 2019) for R software and python scripts that have been developed purposely for this research. Both the NAO
and the linear regression analysis were also computed with s2dverification/s2dv.

## 2.4 Forecast verification

To evaluate the forecast quality of the models we used the anomaly correlation coefficient (ACC). The statistical significance
of ACC differences was assessed following the methodology proposed by Siegert et al. (2017), a statistical test developed for
cases where competing forecasting systems are strongly correlated with one another.

An important aspect to consider when comparing predictive skill between a DCPP experiment and its HIST counterpart
is the selection of a common period for forecast evaluation, to ensure that differences in skill only arise from the effect of
165 initialization (as prediction skill can be sensitive to the evaluation period). Our evaluation period is fixed and starts in 1970 -
the first year for which the DCPP ensemble provides predictions for the full forecast range (1st to 10th year) - and finishes in
2014, which is the last year covered by the HIST ensemble. Linear trends in our analysis were also computed for this same
period.

Not all models in this analysis were initialised in the same month. One model was initialised in the first of October (Nor-
170 CPM1), several in the first of November (CMCC-CM2-SR5, EC-Earth3, HadGEM3-GC31-MM, MPI-ESM1-2-HR and MRI-
ESM2-0) and the others in the first of January (CanESM5 and IPSL-CM6A-LR). Therefore, for practical reasons, in all models
we computed all forecast years (FY1-10) January through December, discarding the first months from those models initialised
in October and November. Additionally, we computed the boreal winter mean (defined from December to February, referred to
as DJF hereafter), which is important for some of the processes and drivers investigated (like the NAO). The forecast winters
were numbered according to their January and February forecast years, which means that, for example, DJF2 refers to the win-
ter that includes the December month of FY1 but months January and February of FY2. We discarded DJF1 from all analyses
since some systems do not fully predict the first winter (as it requires December of FY0).

## 3 Results

### 3.1 Multi-model OHC700 skill assessment

We first evaluate the ACC for OHC700 in all the prediction systems for three different forecast times (years 2, 5 and 10),
as well as in all the historical ensembles. Overall, all decadal prediction systems show positive correlations for most of the
NA at all the different forecast ranges (Figure 1, columns 1-3), with higher correlations typically taking place in the Labrador
Sea and along the Eastern flank of the basin, and negative correlations developing in the Central part of the Subpolar North
Atlantic (CSPNA; approx. 40-55 ºN and 40-25 ºW, with small regional differences across models). A similar pattern is found



for HIST, although with a tendency for more widespread negative correlations and lower positive correlation values than in the predictions.

We now turn our attention to some specific cases of distinct individual model behaviour. While in most models ACC values tend to be highest in FY2 (column 1) and usually decrease as the FY progresses, as expected due to the effect of initialization, this is not the case for IPSL-CM6A-LR and CanESM5. In both models ACC is higher in FY5 and even FY10 than in FY2

over the Labrador Sea (LS) and the CSPNA. This initial skill loss might be caused by a strong initialization adjustment, as their historical ensembles show comparatively higher ACC values than the DCPP at FY2. Another system showing a rapid loss of skill in the LS and CSPNA is NorCPM1, where negative correlations emerge by FY5 and FY10. In this case, it seems to reflect a deficiency in the representation of the forced signals, which could be related to a reported problem in the transient land use specification in North America, with downstream impacts in the Subpolar North Atlantic area (Bethke et al., 2021; Passos

et al., 2023). Its HIST ensemble has a large area of negative skill values over the LS and its surroundings. Interestingly, in this same region NorCPM1 predictions show the highest levels of skill at FY2, which suggests that initialization can temporarily correct the errors in the land-use forcing.

Figure 2 describes the inter-model differences in ACC shown in Figure 1, as diagnosed by the standard deviation of the ACC values across models. The HIST experiments have higher ACC spread than the DCPP experiments over most of the NA at all

200 forecast years. The largest standard deviation values (and thereby inter-model differences), of up to 0.6, are found for HIST along the eastern flank of the NA and in the LS. In the DCPP experiments, the inter-model spread tends to change with forecast year, without much spatial consistency in terms of the regions with the largest standard deviations. The LS (red rectangle in Figure 2) emerges as a region in which inter-model differences in skill are prominent at all forecast times. We also note that the LS is a region in which observationally-constrained datasets largely agree (Figure A1) and thereby where skill is expected to

205 be more accurately diagnosed (as opposed to the CSPNA region, where the observationally based products disagree the most and, therefore, where the skill scores are uncertain and have to be interpreted carefully). Thus, for the rest of this study we will focus on the LS as a region of interest to understand the inter-model differences.

### 3.2 Role of forcings and long-term trends in Labrador Sea OHC skill

To better understand the differences in skill, the predicted and the observed evolution of LS OHC700 anomalies is shown

in Figure 3. At FY2 and FY5, all systems but CanESM5 predict reasonably well the observed evolution, characterized by a very weak cooling trend until the mid-90s that a warming starts unfolding. CanESM5's long-term trend is characterized by a cooling, completely failing to represent the observed LS multi-annual variability as well as the long-term trend. This could be related to the use of ORAS5 for initialization (Sospedra-Alfonso et al., 2021), which has been reported to have non-stationary trends in the region (Tietsche et al., 2020). At longer forecast times (FY10), all models but CanESM5 and NorCPM1 simulate

a long-term warming trend, with inter-model differences mostly affecting the multi-annual modulations around the trend.

Interestingly, in the HIST ensemble only two models, IPSL-CM6A-LR and CanESM5, simulate a clear warming trend consistent with the observed one. The other models show a rather flat evolution and NorCMP1 shows a cooling trend. We also





note that none of the HIST ensembles simulate the cooling until the mid-90s nor the subsequent rapid warming that were partly captured by the DCPP experiments, supporting a key role of initialization in the decadal variability around the trend.

Figure 4 shows that the relationship between the OHC700 trends (as derived for the period 1970-2014, see Section 2.4) and the OHC700 skill in the LS is largely linear across models. In other words, models with stronger OHC700 trends in the LS tend to have higher OHC700 skill in this region, which is particularly evident in the HIST ensemble and the first forecast years of the DCPP ensemble, although with some notable differences. While for HIST, all the models show a wide range in the magnitudes of the simulated trends, in the first forecast years of the DCPP ensemble all models predict similar trends to the

observed one, except for CanESM5 which has been previously mentioned as an outlier. This clear correction of the predicted trend via initialization can be attributed to: (1) an unforced origin for the observed LS trend; and (2) an improved representation of the forced response with more realistic background climate conditions. Interestingly, predicting an accurate trend does not always lead to high levels of OHC700 skill, as noted for, e.g., IPSL-CM6A-LR at FY2 in Figure 4.

To further investigate the impact of the long-term trends on the OHC700 skill in the LS, Figure 5 portrays the ACC values as

a function of FY when both DCPP and the observed data are linearly detrended (dashed blue line), and compares them with the skill for the original time-series (solid blue line). In all models, except for CanESM5, the forecast skill systematically decreases when the trend is removed, even though the drop in skill is not always significant with respect to the undetrended ACC values (red crosses in Figure 5). This confirms that an important part of the skill comes from the representation of the trend. Figure 5 also shows the forecast skill of the HIST ensemble, which compared with the DCPP skill can inform us about the predictive

role of the forcings. The results are largely model-dependent. In some systems, HIST and DCPP have similar ACC values that are only significantly different in the first FYs, which suggests a predominantly forced origin of the skill. Other models, like MRI-ESM2-0, HadGEM3-GC31-MM and CMCC-CM2-SR5, show high and significant ACC values for DCPP, while for HIST the ACC values are indistinguishable from zero. The interpretation of these latter systems is more complex, as the high predictive value of initialization could imply that internal variability is the dominant factor leading to the OHC700 skill, but

it is also possible that the corresponding HIST ensembles simulate an unrealistic externally forced variability that is largely corrected via initialization.

Figure 5 thus illustrates how the large uncertainties in the representation of the forced signals, together with the initialization shocks in some of the systems (CanESM5, EC-Earth3, IPSL-CM6A-LR) prevent us from learning about the true origin of the LS OHC700 predictability. The underlying problem is that we do not know how much of the observed variability is actually

driven by the forcings. In the next two sections (3.3 and 3.4) we will explore (i) how different precursors and drivers of LS decadal variability are represented across models and experiments, to ultimately investigate (ii) whether they can explain some of the inter-model differences in the forced LS OHC700 predictive skill.

## 3.3 Evaluation of main preconditioners and drivers of LS OHC700 variability across models

In this section we explore the underlying differences across models of two important factors controlling Labrador Sea tem-

perature variability: (1) LS stratification and (2) the surface atmospheric forcing. The former is a preconditioning factor for the occurrence of deep convection in the region, whereas the latter is a direct driver of convection and OHC variability via its



influence on local air-sea heat fluxes. We will evaluate how models simulate these important processes, and whether they are improved via initialisation.

### 3.3.1 The preconditioning role of density stratification

It is well known that the LS is an important region where oceanic processes, such as deep ocean convection, can drive large-scale ocean temperature changes (Robson et al., 2016; Ortega et al., 2021). It is, however, less clear if these processes influence the local OHC skill and ultimately their forecast skill, or if OHC persistence is the dominant factor (Buckley et al., 2019). Some prediction systems, like the one based on EC-Earth3, show high OHC predictive skill in the NA even after LS convection collapses due to initialization effects (Bilbao et al., 2021), which suggests that other processes besides the local deep mixing
might also be relevant.

In the LS, deep convection takes place in winter (Yashayaev and Loder, 2016), when the local cooling exerted by the atmosphere can be strong enough to overcome the local density stratification, which acts as a preconditioner. Important model biases in density stratification can therefore potentially mitigate and even suppress deep ocean convection and in this way limit the forecast skill, especially in anomaly initialised systems in which potential model biases are not corrected during
initialization.

Figure 6 shows the climatological wintertime (DJF) potential density profiles for the LS area 6) in DCPP and HIST. The HIST panel, which describes the intrinsic mean-model biases, shows that IPSL-CM6A-LR, EC-Earth3 and especially CanESM5 have overly stratified LS densities, as compared to EN4, while NorCPM1 stands out as a model with virtually no LS density stratification. These are two opposite problems that interestingly seem to arise from biases in the salinity profile (bottom panel
of Figure A2). When looking into the DCPP experiments, NorCPM1 still shows the overly weak LS stratification. In contrast, full-field initialization seems to efficiently correct the strong stratification problems, especially in CanESM5, although in EC-Earth3 stratification is degraded in DJF10 compared to HIST, likely due to the initialization shock reported in Bilbao et al. (2021).

We now revisit the potential density profiles but focusing on how differently the models represent the temporal variations at
275 different levels, as these can reveal other important model biases affecting the vertical mixing. NorCPM1 and IPSL-CM6A-LR portray the largest differences with respect to EN4 for both DCPP and HIST (bottom panel of Figure 6), in particular near the surface where the variance is higher due to the exchanges with the atmosphere. NorCPM1 shows substantially weaker variability at the surface, while IPSL-CM6A-LR shows the largest variability. This might derive from their radically different mean winter stratification (top panel of Figure 6): In NorCPM1 the very weak stratification ensures a rather sustained mixing, which
damps the year-to-year variability. In contrast, in IPSL-CM6A-LR, stratification is relatively strong, favoring a much too intermittent mixing. There is no clear benefit from initialization in the variability profiles for most models. In fact, initializing the models seems to worsen the density variability for CanESM5 in DJF2-DJF5, which can be again linked to the non-stationarity errors inherited from ORAS5.

In the next subsection we will explore whether the differences in stratification can condition the local forcing from the
285 atmosphere.





### 3.3.2 The North Atlantic Oscillation as a key driver of LS variability

Many studies have highlighted the key driving role of the NAO on the interannual variability of LS temperature, salinity and convection (e.g., Eden and Jung, 2001; Guemas and Salas, 2008; Ortega et al., 2012; Yashayaev and Loder, 2016), and through it, on the AMOC, but, to our knowledge, no study to date has explored whether and how structural model differences in the
290 representation of the NAO affect the local air-sea heat exchanges.

Figures 7 and 8 show the NAO pattern (defined as the first EOF of sea level pressure in DJF) for the HIST and the DCPP experiments, respectively. As expected, the low-pressure system (also known as Icelandic Low; IL) tends to be centred around Iceland, and the high-pressure system (also called Azores High; AH) is centred between Azores and the western border of the Iberian Peninsula. There are some notable differences across models and experiments. In the HIST experiments (Figure 7)
both the AH and IL show substantial variations in their location across models and members. Overall, the AH tends to be more located near the Azores Archipelago, although some individual model members, including all NorCPM1 ones, develop their maxima near the Iberian Peninsula.

Important differences across models are also found in terms of the IL location for both experiments. CanESM5 (more obvious in HIST), CMCC-CM2-SR5 and NorCPM1 tend to have the IL located further to the East (i.e., over the Norwegian
Sea and Scandinavia), much like the NAO structure of ERA5 for the study period of 1970-2014 (Figure A4). The other models have their centres of action over Iceland and Greenland, which is more in line with the traditional NAO definition (Hurrell, 1995).

There seems to be an overall agreement between the NAO patterns in the HIST and DCPP ensembles (Figure 8), where the centres of action appear to be unaffected by the forecast drift. This also implies that full-field initialization does not correct the
305 position of the simulated centres of action in models, like CMCC-CM2-SR5, and CanESM5, that simulated them too far to the east.

The relative position between the AH and IL centres of action can critically condition how the NAO affects the surface winds, whose speed is proportional to the local gradient in sea level pressure. This can be crucial in the Labrador Sea, where the surface winds promote deep ocean convection by cooling the surface. Models like NorCPM1 or CMCC-CM2-SR5, in
which both centres of action are placed far from the Labrador Sea, shifting the maximum sea level pressure towards the east, might therefore induce a weaker local forcing. We now investigate whether this is the case by computing the linear regression of the NAO index with the surface heat fluxes (Figure 9), as represented by the CMIP6 variable *hfds*. In the HIST experiments (rightmost column), all models show that the NAO exerts a strong cooling in the Labrador and Irminger Seas, except in CanESM5, where both regions are unrealistically covered by sea ice (Figure A5). Interestingly, the HIST panel additionally
suggests that having the IL centre displaced to the east, like for NorCPM1 and CMCC-CM2-SR5, does not necessarily lead to a lack of surface forcing in the LS. This result suggests that other factors influencing the local heat loss are at play.

For the DCPP ensemble, the regression maps (Figure 9) show a clear beneficial effect of initialization in the representation of the NAO's surface forcing, especially over the LS. In that region, all models show a more consistent picture at DJF2, and a better agreement with the equivalent regressions in ERA5 (Figure A4) suggesting that having more realistic conditions in





stratification, sea-ice or both does help to improve the NAO forcing in the region, even in the cases in which its centers of actions were displaced compared to observations (e.g CMCC-CM2-SR5). As the forecast time progresses, differences start to emerge, particularly in the systems that are full-field initialised (e.g., CanESM5, EC-Earth3, CMCC-CM2-SR5) following the development of the intrinsic model biases. Indeed, full-field initialization not only helps to simulate a more realistic forcing of the NAO, it also critically improves the climatological surface heat fluxes in winter (Figure A3 compared to Figure A4), mean-

state improvements that are very clear in the LS in DJF2 for CanESM5, CMCC-CM2-SR5 and EC-Earth3. These improvements are less noticeable for HadGEM3-GC31-MM, which is also full-field initialized, because this model had a more realistic background mean state density stratification, as evidenced in Figure 6 for its HIST run. Figure A3 also distinguishes NorCPM1 as a model with overly large climatological heat loses into the atmosphere in both the LS and the CSPNA as compared to the other models and ERA5.

To help identify the specific regions where the NAO introduces larger differences across models in terms of local surface heat fluxes, Figure 10 (top row) shows the standard deviation in model space for the regression coefficients shown in Figure 9. It clearly illustrates that the major differences occur over the LS, especially on its western side, thus supporting that the representation of the NAO and its forcing may contribute to the differences in OHC700 skill across models (Figure 2). This is true for both sets of experiments, although the area of high standard deviation values is larger in HIST. The differences across

models are reduced with initialization and become more prominent in the LS as forecast time progresses, and by DJF10 they remain geographically more confined than for the HIST experiment.

    Because winter heat fluxes are not exclusively linked to the NAO, the differences across models in terms of climatological winter surface heat fluxes alone are plotted in Figure 10's middle row. Strong multi-model differences are also evident, with much higher standard deviation values. These are not only limited to the LS, which clearly stands out as the region with the

highest inter-model spread, but are also quite large over the CSPNA.

    Considering that sea ice can act as a barrier that shields the ocean from the atmospheric influence, and in this way condition the climatological heat fluxes, the inter-model spread of the winter sea-ice concentration (when it reaches its maximum extent) is also presented in the bottom row of Figure 10. While in HIST and DJF10 there is a large spread in LS climatological sea ice, which can be mostly associated to CanESM5 and EC-Earth3 (Figure A5), in the forecast winters DJF2 and DJF5 the

differences are confined to a narrow band at the westernmost side of the LS. This is also where the surface heat flux regressions onto the NAO and the climatological surface heat fluxes showed the largest intermodel spread. Therefore, it would seem that all the three model properties are intricately related in that region.

## 3.4     Understanding uncertainties in the externally forced LS OHC700 variability and predictability

    This last section seeks to narrow down the large uncertainties identified in the LS OHC700 externally forced signal, by con-

trasting the HIST simulations against observationally-based values for the previously analysed key physical properties (i.e., stratification, NAO regression, surface heat fluxes and sea ice concentration). To this end, a set of scatter plots was assembled in Figure 11.



We find a strong linear relationship between the forced OHC700 skill and the stratification index, with stronger stratifica-
tion linked to higher skill. A possible interpretation of this linear relationship is that stronger mean stratification limits the
occurrence of deep convection events, especially those triggered by internal climate variability processes (which in HIST runs
cannot be in phase with the observations), allowing for a better capture of the long-term trends. However, it is important to
note that the models with the largest forced OHC700 skill also largely overestimate the local stratification when compared to
an observational reference, which raises questions about their realism.

The relationship between the OHC700 skill and the climatological winter surface heat fluxes (*hfds*) in the LS (panel *c)* in
Figure 11) is also highly linear. In models that have stronger climatological surface heat fluxes in the LS (i.e., that lose more
heat to the atmosphere), the forced signal explains a lower percentage of the observed OHC700 variance (as implied by the
lower skill score). One potential explanation for this relationship is that with stronger heat fluxes the local stratification can be
overcome more easily, which therefore allows for a higher presence of spurious unforced signals that degrade the agreement
with the observations (i.e., lower the ACC value). Interestingly, even though higher ACC values are linked to weaker surface
heat forcings, the observed ERA-5 climatologies suggest that the models with the highest forced skill are not particularly
realistic. It should be noted, however, that ERA-5 does not include all of the air-sea ice fluxes that are included in the HIST
fluxes, and might be important at the western side of the Labrador Sea.

For the two other preconditioners identified in section 3.3.2 (i.e., the local NAO surface forcing and the climatological sea
ice conditions) we do not find a clear linear relationship with the forced OHC700 LS skill. Despite this, CanESM (the model
with the highest ACC values) is identified in both cases as a clear outlier when compared with the observational references.

All the above results thus suggest that high ACC values in some of the historical ensembles are not necessarily indicative of
good model performance. The underlying issue is that the true split between the forced and the internally generated variability
in the real world is unknown, which hinders the identification of the models that simulate the forced signal better. Interestingly,
the observational references included in the scatter plots are generally close to the multi-model mean value, supporting its
standard use to derive our best estimate of the real forced signal. This multi-model mean has a forced OHC700 skill in the
LS of 0.5, which would imply a significant but not dominant contribution of the external forcings to Labrador Sea OHC700
variability (as it would explain 25% of the total variance).

## 4   Conclusions and Final Remarks

In this study, the predictive skill of the North Atlantic upper Ocean Heat Content has been explored in a multi-model context,
using CMIP6 ensembles of historical and decadal climate prediction experiments from eight different models. By analysing
both ensembles of experiments it has been possible to investigate how and to what extent the external forcings contribute to
the regional predictability of the OHC, assessing also the benefits of initialization. The bulk of the analysis has been delimited
to the Labrador Sea region, where observational uncertainties are relatively low but important skill differences across models
were found. To further understand these inter-model differences, we have explored whether they can be linked to the capability





of the underlying models to represent key ocean-atmosphere processes and properties that are tightly connected to the local
OHC700 variability, such as the preconditioning role of density stratification and the NAO influence on the surface heat fluxes.

The main findings of the paper are summarized as follow:

– Initialised decadal predictions largely agree on the regions with high predictive capacity for the OHC, which mostly
concentrate on the Labrador Sea region and the eastern flank of the North Atlantic. All of them also show a region with
negative skill scores: the centre of the Subpolar North Atlantic, albeit with important differences regarding the exact
location and extension of the negative ACC values, which largely varies across the models and experiment types. It is
unclear how much of this low skill is due to the large local observational uncertainties. From these three regions, the
largest inter-model differences occur in the Labrador Sea, where some models experience initial shocks, as identified by
Polkova et al. (2023), degrading the skill some years after initialization.

– In the Labrador Sea region, no clear picture emerges from the multi-model ensemble of how much predictive capacity
for the OHC arises from external forcings, as large inter-model differences in ACC are found for the OHC of the HIST
experiments. The added predictive value of initialisation, determined as the difference in skill between DCPP and HIST
ensembles, is also highly variable across models. This model-dependence of the results highlights the importance of
using multi-model approaches, as analyses focused on individual models, like the one in Carmo-Costa et al. (2021), can
potentially lead to misleading generalizations.

– In the HIST experiments, we have identified a strong linear relationship between the skill for the Labrador Sea OHC and
the local density stratification, as well as a strong inverse linear relationship between the same skill and the climatological
local surface heat fluxes. Since both a stronger stratification and weaker surface heat fluxes have a weakening effect on
the vertical mixing, we interpret that models with higher skill for the OHC are models where deep mixing only occurs
sporadically, reducing the effects of spurious signals arising from internal variability that tend to lower the correlation
values.

– The HIST experiments with higher ACC for the Labrador Sea OHC also have larger biases in the mean state stratification
and heat fluxes, which questions their realism. The multi-model mean of the HIST experiments compares particularly
well with observations, and is likely to provide a more realistic estimate of the predictability attributable to the forcings,
which according to the multi-model mean would account for 25% of the total OHC variance in the Labrador Sea.

– Our multi-model DCPP ensemble includes 4 systems using anomaly initialization, and 4 systems using full-field initial-
ization, which has allowed us to assess their relative merits. We have found that, overall, full-field initialization helps
improving the representation of the selected key mean model features in the first forecast years, including the surface
forcing from the NAO and the winter sea ice, but it does not necessarily lead to systematic improvements in Labrador
Sea OHC skill, as already found in previous studies for the North Atlantic (Hazeleger et al., 2013; Volpi et al., 2017). No
systematic benefit of anomaly initialization has been identified either, although for the case of NorCPM1 we have found



significant OHC skill along the full forecast, despite an overly weak mean stratification and some reported local errors in the forced signals.

- Regarding the impact of enhancing the resolution, the two systems based on eddy-permitting oceans (HadGEM3-GC31-MM and MPI-ESM1-2-HR) are within the best performing ones in terms of upper OHC skill in the Labrador Sea and also in the whole North Atlantic, although other coarser models like CMCC-CM2-SR5, EC-Earth3 and MRI-ESM2-0 show very similar skill values at all FYs. This means that no particular benefit is obtained from the larger computing costs incurred by these higher resolutions, at least not for the subpolar latitudes in these DCPP experiments. It is possible, however, that the benefits of the higher resolution have been partly masked by the reduced ensemble sizes available for those models, as it has been previously shown that a larger ensemble size has a positive impact on the North Atlantic skill (Delgado-Torres et al., 2022; Athanasiadis et al., 2020).

This study has linked the differences in upper OHC skill to different mean state biases across models, providing insights in relevant aspects of model fidelity that can be considered to guide the development phase of future climate prediction systems. A deeper fundamental understanding of the key sources of OHC predictive skill could be achieved by performing more holistic approaches, including heat budget analyses and the investigation of advective processes (similar to the propagation mechananisms described in Langehaug et al., 2022), which are beyond the scope of this study. Novel approaches are also needed to cleanly disentangle the relative contributions from external forcings and internal variability to the predictive skill, as the added predictive value of initialization can arise from internal processes as well as a better representation of the forced response.

*Code availability.* All code developed for this study was based on ESMValTool, CDO, R or python. Scripts can be made available by the main author upon reasonable request.

*Data availability.* The CMIP6 simulations are available through the Earth System Grid Federation (https://esgf-data.dkrz.de/projects/esgf-dkrz/). EN4 version 4.2.2 can be found at www.metoffice.gov.uk/hadobs/en4/download-en4-2-2.html; we used the analyses files produced with the bias correction method from Gouretski and Reseghetti (2010). ERA5 can be downloaded from https://cds.climate.copernicus.eu/cdsapp#!/dataset/reanalysis-era5-complete?tab=form and HadISST version 2.2.0.0 is available at https://www.metoffice.gov.uk/hadobs/hadisst2/.





**Table 1.** Summary of key features in the AOGCMs used in this study.

| Dataset Name: | CanESM5 | CMCC-CM2-SR5 | EC-Earth3 | HadGEM3-GC31-MM | IPSL-CM6A-LR | MPI-ESM1-2-HR | MRI-ESM2-0 | NorCPM1 |
|---|---|---|---|---|---|---|---|---|
| Ocean Model | CanNEMO | NEMO 3.6 | NEMO 3.6 | Global Ocean 6 | NEMO 3.6 | MPIOM 1.6.2 | MRI.COM 4 | BLOM |
| Ocean Grid | ORCA1 | ORCA1 | ORCA1 | ORCA025 | ORCA1 | TP04 | (tripolar) | NCAR's gx1v6 |
| Horizontal Resolution | 1° | 1° | 1° | 0.25° | 1° | 0.4° | 1x0.5° | 1° |
| Vertical Levels | 45 | 50 | 75 | 75 | 75 | 40 | 61 | 53 |
| Ocean Initialization | ERSSTv3, OISST (SST), ORAS5 (T,S) | CHOR, CGLORSv7 reanalyses | ORAS5 (SST,SSS), EN4 (T,S) | Met Office Statistical Reanalysis (T,S) | ERSSTv3 (SST), Friedman et al. (2017) (SSS) | ORAS4 (T,S) | Ishii et al. (2017) (T,S) (down to 3000 m) | HadISST2.1, OISSTv2 (SST), EN4 (T,S) |
| Initialization Method | Full field (via nudging) | Full field | Full field (via nudging) | Full field (via nudging) | Anomaly (via nudging) | Anomaly (via nudging) | Anomaly (via IAU[a]) | Anomaly (by EnKF assimilation) |
| References | (Swart et al., 2019) | (Cherchi et al., 2019; Nicolì et al., 2022) | (Bilbao et al., 2021; Döscher et al., 2021) | (Sellar et al., 2020; Kay et al., 2022) | (Bonnet et al., 2021) | (Müller et al., 2018; Li et al., 2019) | (Yukimoto et al., 2019) | (Bethke et al., 2021) |
| HIST members | r(1-10)i1p2f1 | r(2-11)i1p2f1 | r(2,7,12,17-22,24)i1p1f1 | r(1-4)i1p1f3 | r(1-10)i1p1f1 | r(1-5)i1p1f1 | r(1-10)i1p1f1 | r(1-10)i1p1f1 |
| DCPP members | r(1-10)i1p2f1 | r(1-10)i1p1f1 | r(1-10)i4p1f1 | r(1-4)i1p1f2 | r(1-10)i1p1f1 | r(1-5)i1p1f1 | r(1-10)i1p1f1 | r(1-10)i1p1f1 |

[a] IAU - Incremental Analysis Updates





**Figure 1.** ACC maps for the OHC700 in the DCPP (forecast years 2, 5 and 10; columns 1-3, respectively) and HIST ensembles (column 4). Stippling indicates cells with correlation values statistically significant at the 95% confidence level. All ACC values are evaluated against EN4 for the period 1970-2014. Each row shows the results for a different model.





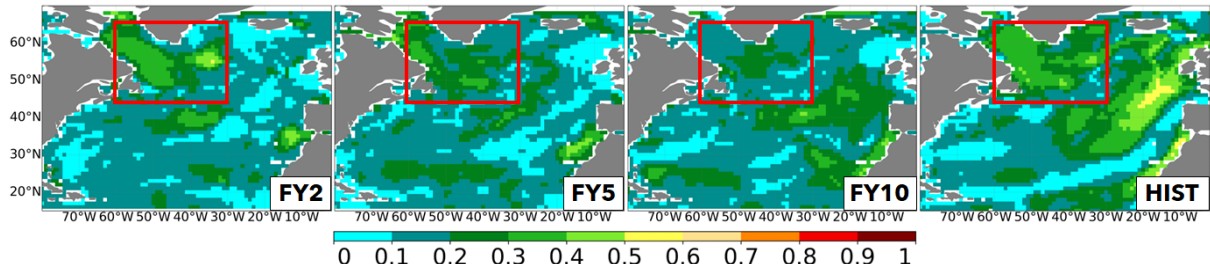

**Figure 2.** Standard deviation across models of the ACC values for OHC700 in DCPP (forecast years 2, 5 and 10; columns 1-3, respectively) and HIST ensembles (column 4). The red box encloses the Labrador Sea region, chosen to compute all area-weighted averaged mentioned hereinafter, with boundary coordinates 60-30ºW and 45-65ºN.

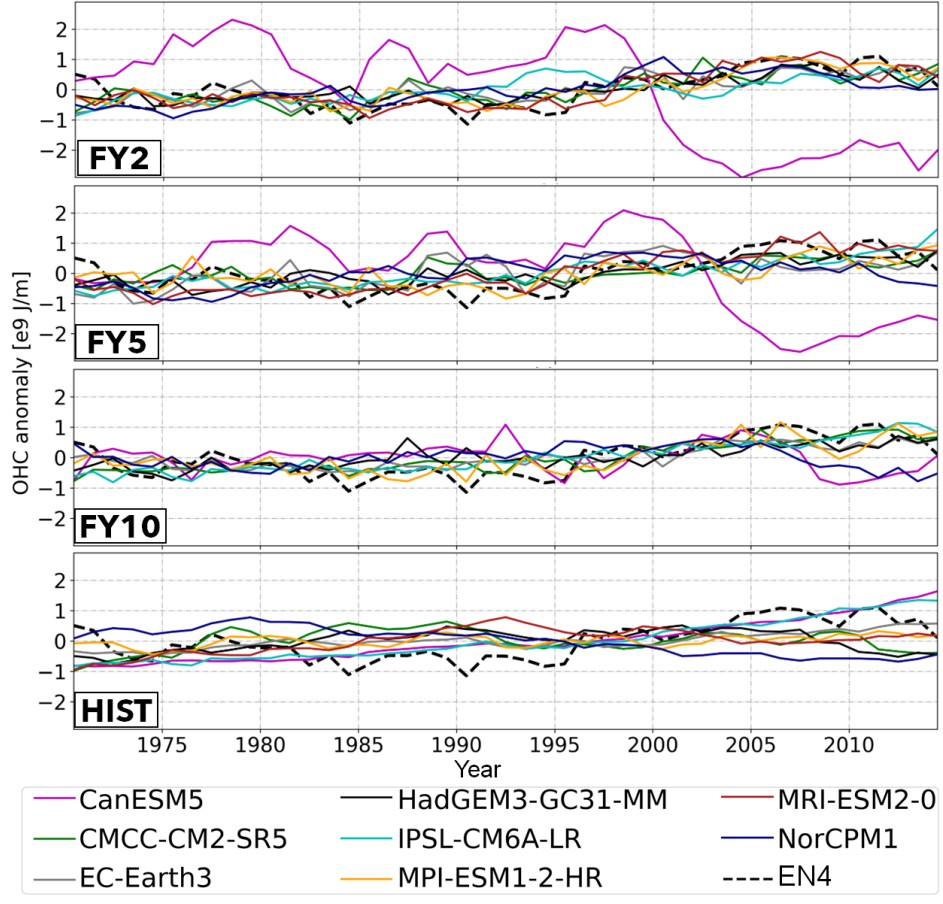

**Figure 3.** Timeseries of the spatially averaged OHC700 anomalies in the Labrador Sea region (red box in Figure 2), fo the DCPP (forecast years 2, 5 and 10) and HIST ensembles. The corresponding time series for EN4 observations is added as a dashed line.



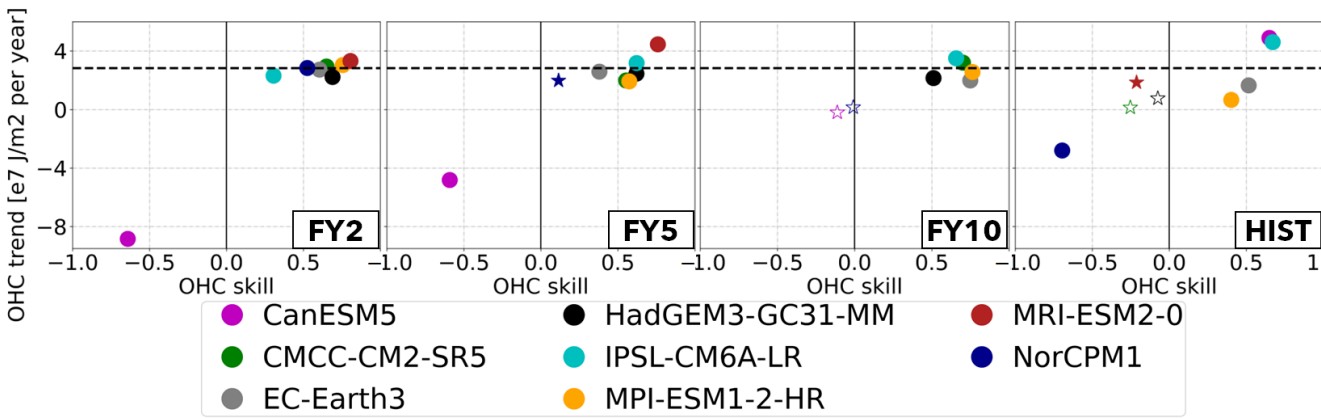

**Figure 4.** Scatter-plot of the relationship between the skill of OHC in the Labrador Sea region (red box in Figure 2) and the local OHC700 trend in both the DCPP (forecast years 2,5 and 10; columns 1-3, respectively) and HIST ensembles (column 4), all based on yearly averages. All trends were computed for the period of interest 1970-2014 (see Section 2.4 for more information). Stars represent non-significant correlation values at the 95% confidence level. Empty symbols represent non-significant trend values, at the 95% confidence level. The dashed black horizontal line represents the trends for EN4 observations.

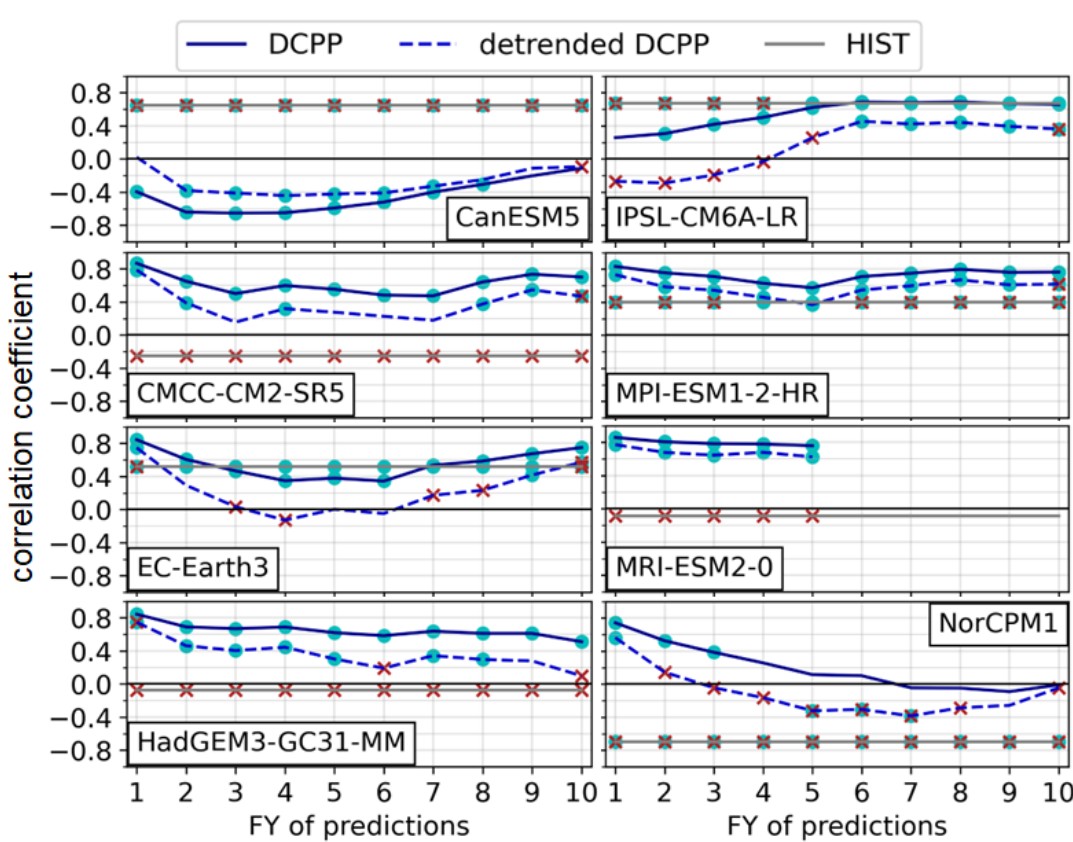

**Figure 5.** ACC of the spatially averaged OHC700 in the Labrador Sea (red box in Figure 2) as a function of FY. Skill values are shown for the DCPP (blue lines) and HIST ensembles (grey lines) and are evaluated against EN4. In DCPP, skill is also computed after detrending both the forecast anomalies and the EN4 anomalies (detrended DCPP; dashed blue lines). Cyan dots indicate ACC values that are significantly different from zero at the 95% confidence level. Red crosses indicate that the HIST or the detrended DCPP ACC values are significantly different from the DCPP values.



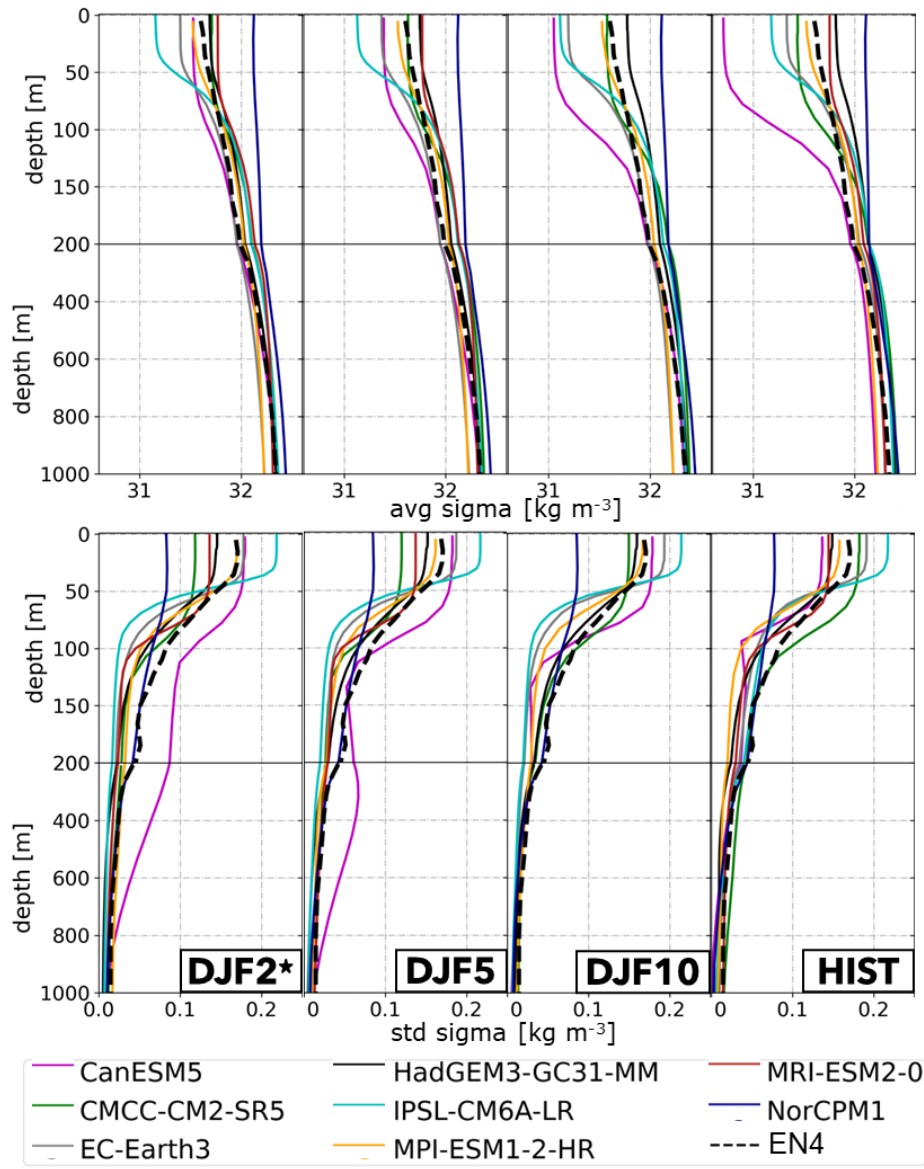

**Figure 6.** (Top) Mean-state climatology of the spatially averaged LS potential density anomaly in DJF (referred to the surface; *sigma*) as a function of depth. The observational reference EN4 is included as a dashed black line. From left to right, it shows the results for the DCPP (in forecast winters DJF2, DJF5 and DJF10) and HIST (in DJF) ensembles over the period 1970-2014. (Bottom) The same as in the top row, but for the standard deviation in time of the spatially averaged LS potential density anomaly.

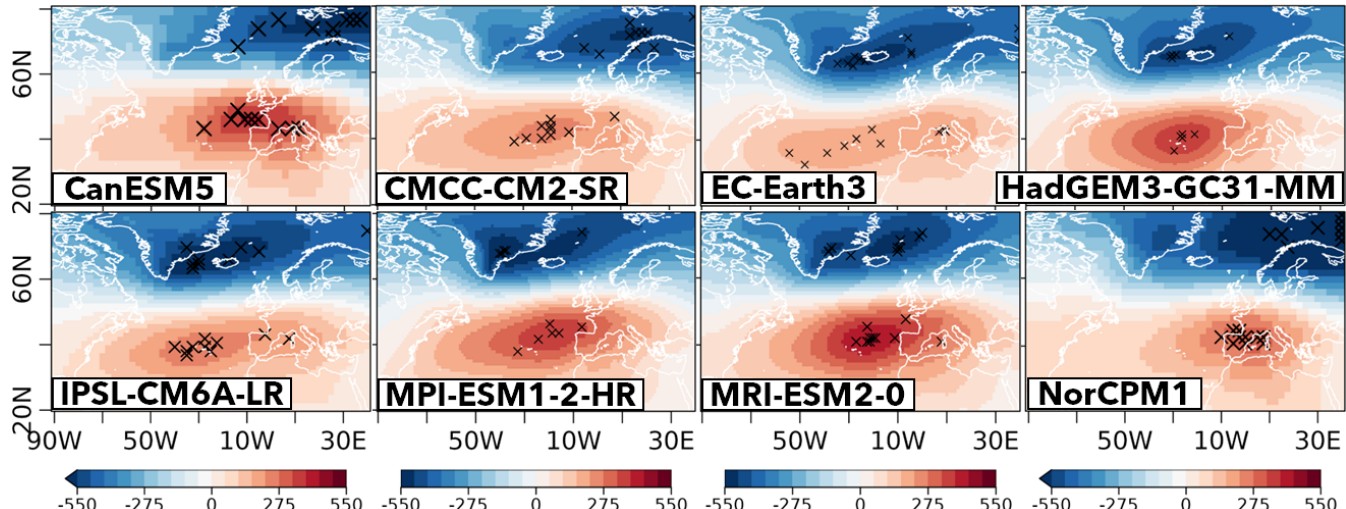

**Figure 7.** Spatial patterns of the NAO (as described by the first EOF of DJF sea level pressure) in the different HIST ensembles. The EOF is computed with all individual model members concatenated in time. Each cross represents the positive and negative centres of action (defined at the place where the NAO pattern attains its maximum and minimum sea level pressure anomalies) when the EOF is computed individually for each member, to thus indicate the intra-model spread.

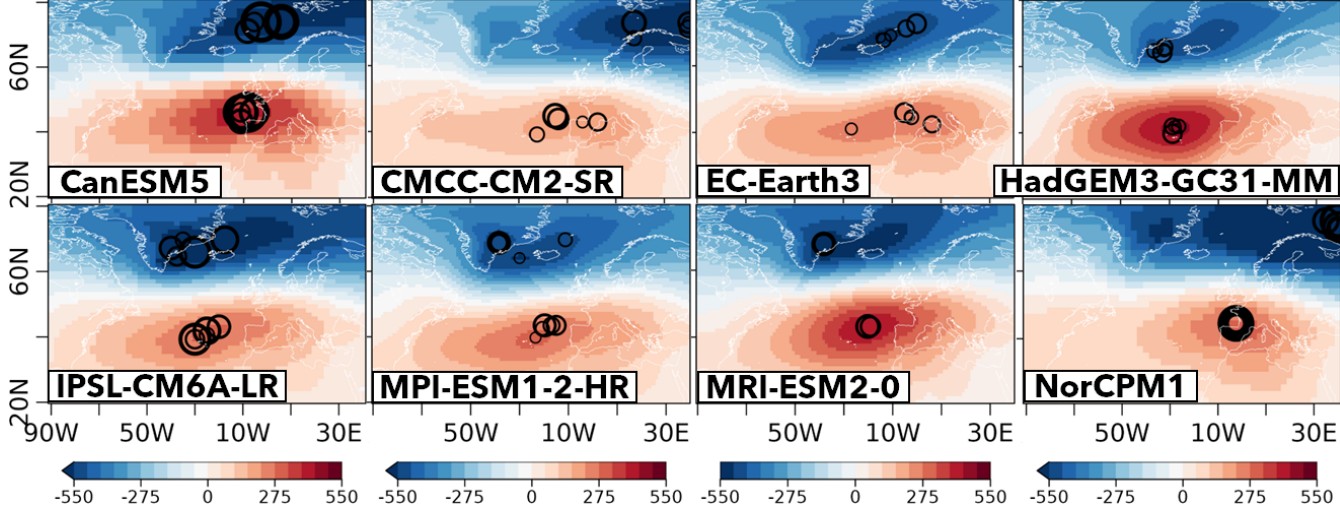

**Figure 8.** Same as in Figure 7 but for DCPP experiments. In this case the colour shading represents the NAO pattern at DJF2 and the circles (in increasing size) represent the centres of action for DJF2, 4, 6, 8 and 10, to thus illustrate any potential shifts with forecast time.



**Figure 9.** Regression maps of the NAO index onto net surface heat fluxes (*hfds*) for the DCPP (in forecast winters DJF2, DJF5 and DJF10; columns 1-3, respectively) and HIST ensembles (in DJF; column 4). Negative (positive) values represent upward (downward) heat fluxes, in Wm$^{-2}$. The contour lines represent the corresponding NAO pattern. The red box in the upper-right plot depicts the LS region used in the analyses.





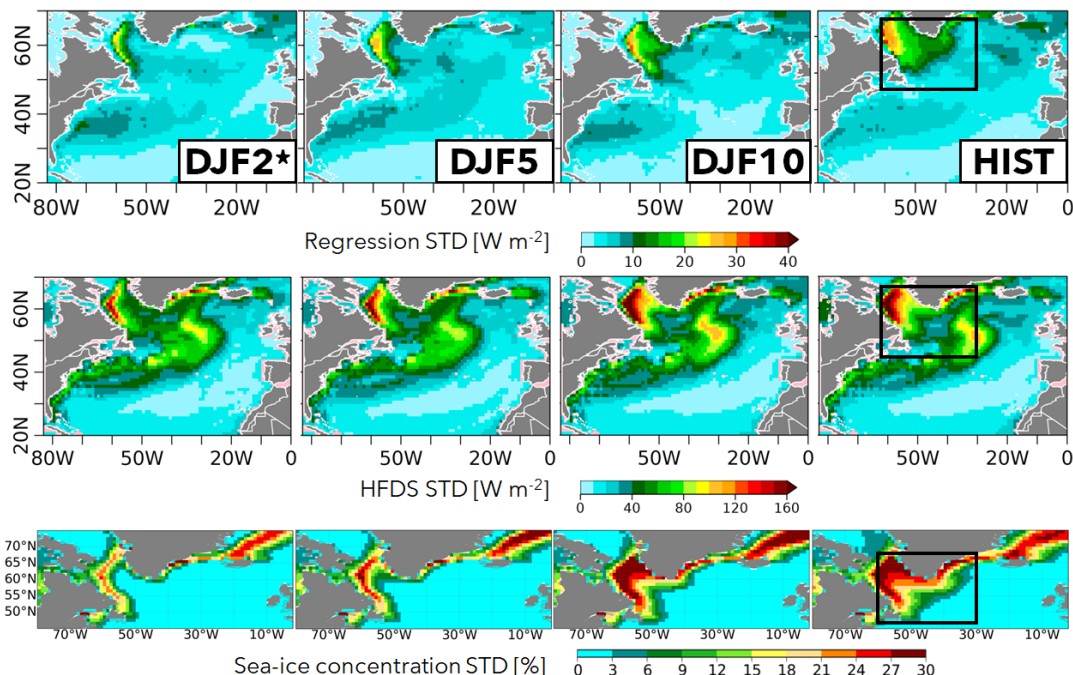

**Figure 10.** (Top) Inter-model spread for the regression coefficients in Figure 9, as defined by the standard deviation across models. (Middle) The same as above but for the climatological net surface heat fluxes in DJF. (Bottom) The same as above but for the climatological sea ice concentration in DJF. The black box in the rightmost column depicts the LS region used in the analyses.





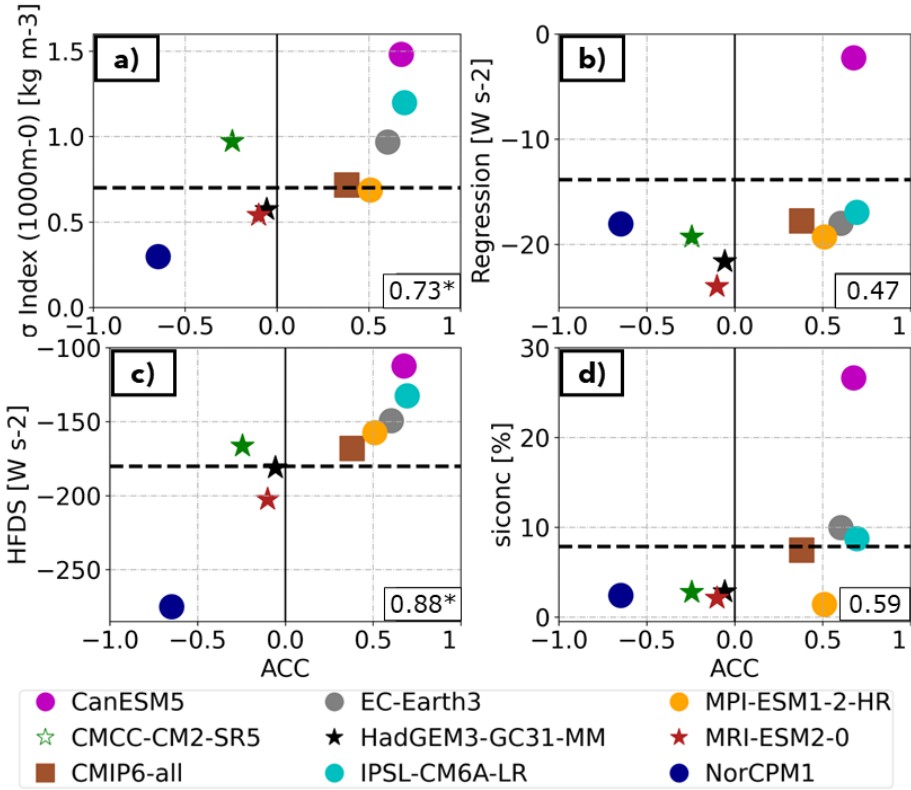

**Figure 11.** (a) Scatterplot of the relationship between the ACC skill in the Labrador Sea OHC700 and the climatological value of index of the Labrador Sea density stratification in the HIST ensembles. The stratification index is computed as the density difference between 1000m and the surface. The 1000m level was chosen based on the visual inspection of the vertical profiles (Figure 6) as a characteristic level of the mean properties of the ocean subsurface. The sensitivity of the results to the index definition was tested by recomputing the index using other levels within the range of 300m and 1000m (not shown), which yielded very similar results and did not affect the overall relationship with OHC700 skill. (b) The same as in *a* but between the ACC skill in the Labrador Sea OHC700 and the regressed values of the NAO index onto the Labrador Sea net surface heat fluxes in DJF. (c) The same as in *a* but between the ACC skill in the Labrador Sea OHC700 and the climatological Labrador Sea net surface heat fluxes in DJF; d) The same as in *a* but between the ACC skill in the Labrador Sea OHC700 and climatological DJF sea-ice concentrations in the Labrador Sea. In all panels stars represent non-significant correlation values at the 95% confidence level. The black dashed horizontal line represents the respective reference dataset: a) EN4; b,c) ERA5; d) HadISST. The linear relationship between the different pairs of metrics is measured with correlation values in the model space, shown for each plot in the lower right corners, with asterisks indicating if the correlation coefficient is significant at the 95% confidence level.



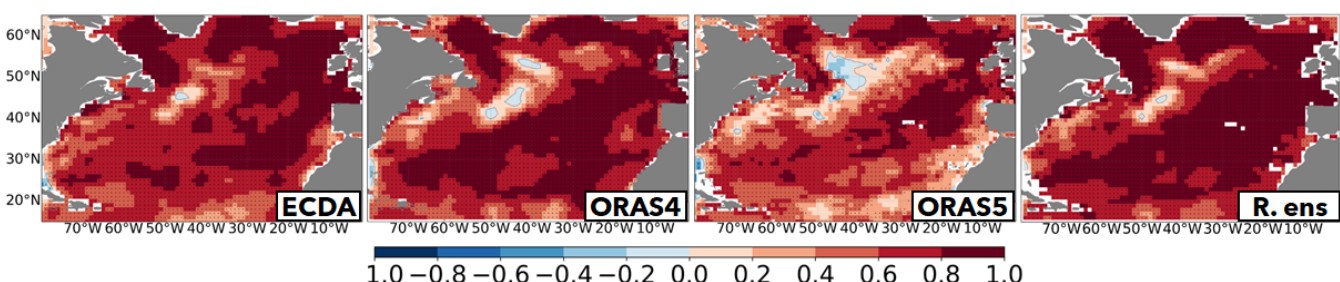

**Figure A1.** Correlation maps between the OHC700 in EN4 and the OHC700 in ECDA, ORAS4, ORAS5 and the multi-reanalyses ensemble mean, computed over the period 1970-2014.



**Figure A2.** As in Figure 6 but for the climatological vertical profiles of potential temperature (top) and salinity (bottom).






**Figure A3.** Climatological DJF net surface heat fluxes for the DCPP (in forecast winters DJF2, DJF5 and DJF10; columns 1-3, respectively) and HIST ensembles (in DJF, column 4). Negative (positive) values represent upward (downward) heat fluxes, in Wm$^{-2}$. The contour lines represent the associated standard deviation in time for the period 1970-2014. The red box in the upper-right plot depicts the LS region used in the analyses.





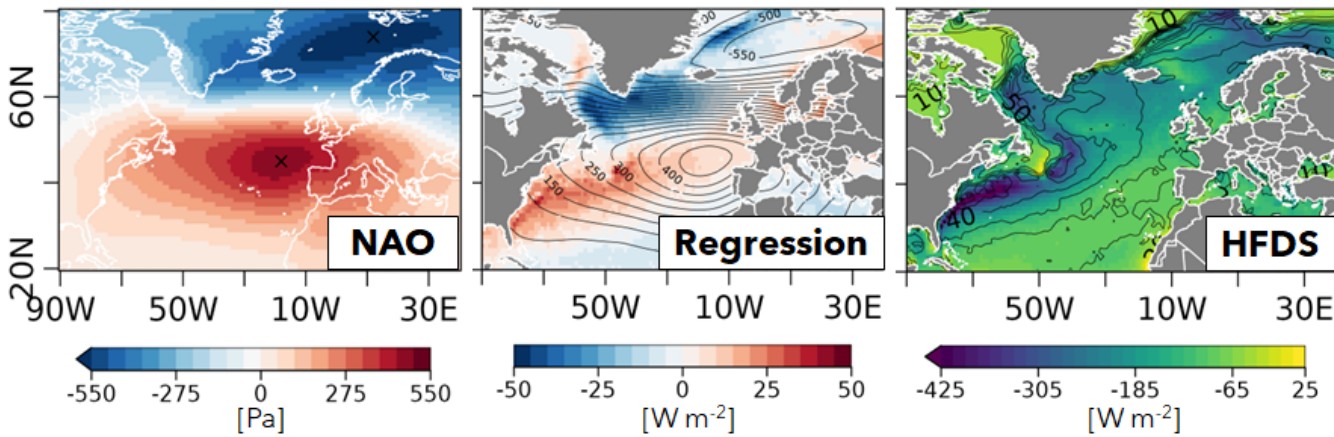

**Figure A4.** (Left) Spatial pattern of the NAO in the ERA5 reanalysis, computed as in Figure 7. (Middle) Regression map of the NAO index onto the DJF net surface heat fluxes in the ERA5 reanalysis. (Right) Climatological DJF net surface fluxes in the ERA5 reanalyses.



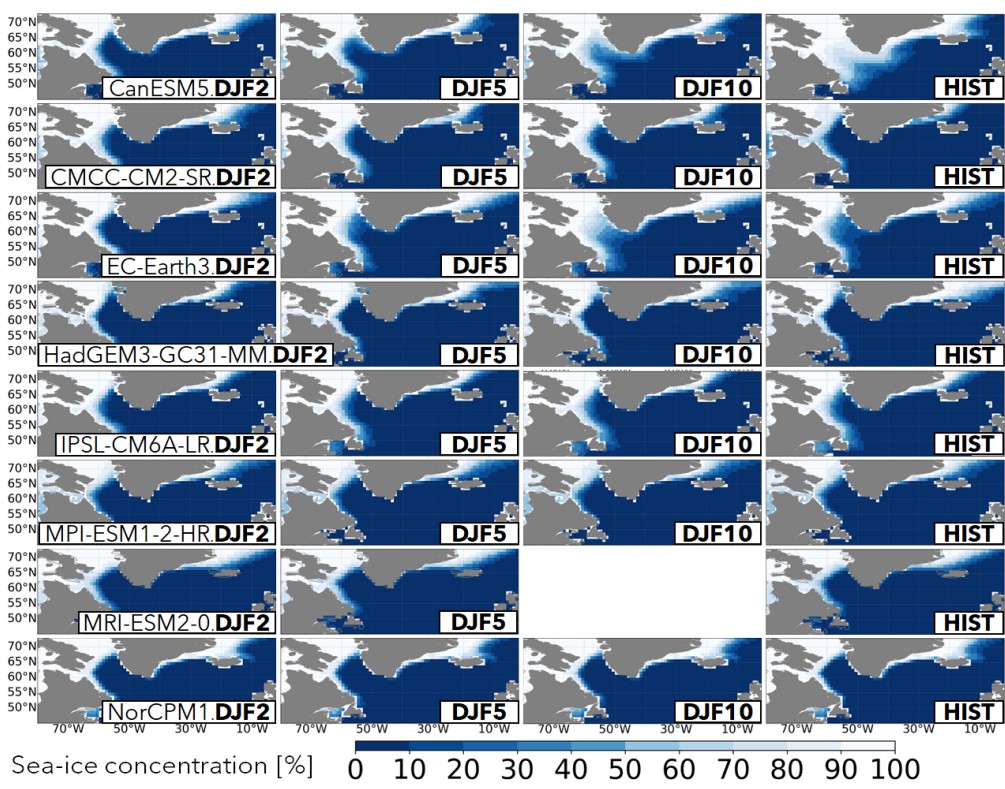

**Figure A5.** Sea ice concentration (*siconc*) maps for the initialized predictions for the forecast winters DJF2, DJF5 and DJF10 (columns 1-3, respectively) as well as DJF in the HIST ensemble (column 4).



*Author contributions.* TCC wrote the original draft with input from all authors, specially RB and PO. TCC wrote the scripts to analyse the data and to plot the figures. All authors contributed to the conceptualization of the study and to the interpretation of the results.

*Competing interests.* The authors have no conflicts of interest to declare that are relevant to the content of this article.

*Acknowledgements.* TCC would like to acknowledge the financial support from the the FCT (Fundação para a Ciência e a Tecnologia)
through projects FCT-UIDB/50019/2020, PD/BD/142785/2018 and COVID/BD/152668/2022. Furthermore, ATM acknowledges SARDINHA2020 (MAR2020) and ROADMAP (JPIOCEANS/ 0001/2019). RB was supported by the European Commission H2020 projects EUCP (Grant no. 776613) and the Horizon Europe Project Impetus4Change (Grant no. 101081555). PO was supported by the Spanish Ministry of Economy, Industry and Competitiveness through the Ramon y Cajal grant RYC-2017-22772. JR was funded by NERC via the WISHBONE (NE/T013516/1), CANARI (NE/W004984/1), and ALPACA (NE/W004984/1) projects, and by UKRI via the EPOC project.



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
