# Peer review of "A multi-model analysis of the decadal prediction skill for the North Atlantic ocean heat content"

_EGUsphere, 2024_

## Community Comment (CC1)

**General comments**

This paper analyses the variations of the ocean heat content averaged over the first 700 m (OHT700) of the ocean into an area surrounding the Labrador Sea region for the period 1970-2014. For this purpose, it uses observational data available and two multi-model sets of climate simulations, one with only the external forcing (historical simulations) and the other with decadal hindcasts starting from observation-based estimate. The analysis shows a very wide range of response in the models, especially for the historical simulations. The authors try to estimate the skill of the different systems to reproduce the OHT700 in decadal prediction and historical simulations, and found an interesting link between this skill and the capability of the models to reproduce observed mean state of stratification and ocean heat fluxes in the Labrador area.

This is an interesting and well-written paper. The analysis led is impressive given how difficult it is to deal with so many climate model data. The interpretation of the results is wise and useful, even though no definitive conclusions can be drawn from this type of multi-model analysis. At least, this is presenting an interesting intercomparison of present-day models to reproduce heat storage in the Labrador Sea area and a few interesting predictors that might be of use for observations constraints approaches.

I therefore think this paper is suitable for publication. I have mainly some comments that might allow to strengthen the demonstrations and possibly improve the interpretation of the results.

- Line 35-40: here the authors are mixing discussions about the subpolar gyre and the wider North Atlantic and ocean heat content and SST. It might be worth to be a bit more specific in the description of those papers.
- Line 61: a reference after forecast range might be useful to support this claim.
- Line 202: The LS, as represented in Figure 2, does not entirely correspond to the Labrador Sea but is going far the east, including the Irminger Sea for instance. In this respect the agreements between observation-based datasets are not that clear to the east (cf. Figure A.1), while the good agreement is taken as a reason to focus on this region in line 204. Please clarify. Have the authors tried a more tied region?
- Line 249-253: ocean stratification and heat fluxes are two variables clearly linked in the convection region. If the halocline is too strong, convection is not allowed and heat fluxes can lead to sea ice formation. It might be worth to state this coupling between these two variables (maybe in the discussion).
- Line 266: it is said line 155 that density is computed with reference 1000 m (sigma_1) while in Figure 6, the caption talks about reference to the surface. Given that the numbers in Figure 6 are larger than 28, I assume this is actually sigma_1. This choice is surprising given that then authors are focusing on the very upper layer. I think it might be better to consider sigma_0 as stated in the caption (while it is not what is shown).
- Line 291-296: why are the observations are not shown on Figure 7?
- Line 395-400: The use of residual ACC (Scaife & Smith 2018) might be interesting as well. I'm wondering if this might work for this type of complex quantity like OHT700,

especially given the complexity of its forced response. A discussion on this aspect might be interesting here I think.

- Line 412-416: I have the feeling that this aspect has not been much depicted in the result section, so that this discussion seems a bit coming out of the blue. Maybe useful to add a few points on this in the results section.
- Line 431: Yes, the omission of advective processes is clearly missing in this paper, but I can understand that it is far from easy to have those quantities from such a large ensemble of simulations. What about citing Ortega et al. (2015) that was also discussing this type of processes in details? + typo at "mechanisms"

**References**

Scaife A.A., Smith D., 2018. A signal-to-noise paradox in climate science. *npj Clim. Atmos. Sci.,* 1, 28. doi: 10.1038/s41612-018-0038-4

Ortega P., et al. (2015) Reconciling two alternative mechanisms behind bidecadal AMOC variability. *Progress in Oceanography* 137, pp. 237-249.

---

## Author Comment (AC1)

Review of the manuscript "A multi-model analysis of the decadal prediction skill for the North Atlantic ocean heat content" by Carmo-Costa et al

The study described the multi-model analysis of the decadal prediction skill for the upper ocean heat content. Authors analyze multi-model spread in prediction skill of the upper 700 m ocean heat content, further choosing the Labrador Sea and its surrounding areas, where they observe the largest model uncertainties among other regions in the North Atlantic. They further investigate the processes to clarify sources of these uncertainties and possibly improve predictions. Overall, the study is a worthy research contribution, however the manuscript needs to be further improved before publication as it currently lacks important details on the methods and description of some results; especially the NAO part is confusing. Also, some figures need improvement, so that one can follow better the conclusions that are based on them. I thus suggest major revision.

We thank the reviewer, Iulia Polkova, for her constructive comments on our manuscript. We greatly appreciate the time and effort she has invested in thoroughly reviewing our work. Her feedback has been invaluable for enhancing the quality and clarity of our paper, and we believe the revisions have significantly strengthened it.

We have meticulously addressed each of her suggestions and have provided a detailed, point-by-point response to all the comments.

Detailed comments are below.

Abstract:

1.  L5-10: In the abstract, authors describe what their study intends to investigate "We analyze eight CMIP6 … to investigate if intrinsic model characteristics … can influence …". Instead of stating the aim, I suggest to summarize results of the study and whether authors' intents were achieved by the current analysis.

    Reply: The abstract has been completely rewritten as requested, with a focus on the actual findings, instead of the main goals.

2.  L10-12: Last sentence only describes the historical ensembles. Basically, this is the only sentence about the results in the abstract part and it has nothing to do with decadal prediction experiments, which should be the main focus of the study according to the title. Given that the title of the paper focuses on decadal prediction skill, does this statement in the last sentence of the abstract also hold for prediction ensembles?

    Reply: The statement only holds for historical simulations, that are the ones showing the largest uncertainties. This last sentence has been substantially changed in the new version of the abstract.

Introduction:

1. L15: The exact "decades" and the exact level that has a trend need to be specified. Moreover, first two papers cited are not about warming trend in observational datasets but about reanalyses, which are not the same thing and thus need to be named correctly. Same in the second sentence (L17-19) – the papers cited are about reanalyses and not about observational datasets. Reanalyses are as much based on the observations as on the models; if they did not introduce their underlying model and assimilation method effects on the final products, we would have more coherence between different reanalyses. Thus, it is not correct to equalize them with the observational datasets.

   Reply: The first paragraph has been completely re-written and those papers are no longer cited. We have also revised the rest of the introduction to ensure that results from reanalysis products are referred to appropriately.

2. L20-23: Previous paragraph described a warming trend in the North Atlantic Ocean Heat Content (NA OHC); this paragraph begins with describing cooling trend but does not specify that the cited papers speak about sea surface temperature (SST) trend in the subpolar North Atlantic (SPNA). The fact that the focus now changed from OHC to SST, needs to be specified or one has to cite the appropriate papers, which describe cooling trend in the SPNA OHC.

   Reply: We now specify that the papers describing the North Atlantic cooling trend have characterized it for sea surface temperatures. .

3. L28: Specify "recent OHC variability" period. Imagine this paper will be read in 20 years from now. Will this current period description still be "recent"?

   Reply: The sentence has been rephrased to avoid this problem.

4. L36: Link to Duchez et al leads to "page not found" and if it is from a newsletter, it might not be an appropriate citation. Links to cited papers need to be double-checked. I came across at least 4 of them (start with https://doi.org/https://doi.org...), which are leading to "page not found".

   Reply: All the citations and their associated links have been revised in the newly submitted version.

5. L58-61: This statement is not accurate. There are multiple studies from the decadal prediction community, which are cited few lines below and which show that the NA SST and the NA SPG are predictable up to decadal timescale and maybe even longer. The NA region is the one that is the most distinctive in its predictability due to initialization among all other regions on the globe. The authors basically cancel this knowledge from previous studies in this paragraph without providing any evidence. To avoid this confusion, one needs to specify the exact regions that are described in the Langehaug et al paper, where prediction systems indeed still have a lot of

troubles, namely the Norwegian Sea, the Inflow region and eastern SPG and not generalize the results of this study to the whole North Atlantic basin. By the way, the link provided to their paper is not working.

Reply: We acknowledge that the current phrasing might be misinterpreted to suggest that models rapidly lose predictive skill for North Atlantic SSTs after the first forecast year, which is incorrect. We have rewritten and streamlined the discussion on predictive skill in the North Atlantic region to ensure accuracy.

6. L64: In this context, the link to Polkova et al study should be a different one, namely: Polkova, I., Brune, S., Kadow, C., Romanova, V., Gollan, G., Baehr, J., et al. (2019). Initialization and ensemble generation for decadal climate predictions: A comparison of different methods. Journal of Advances in Modeling Earth Systems, 11, 149–172. https://doi.org/10.1029/2018MS001439 This is where Polkova et al 2019 analyzed the NA SST and NA OHC skill for an "individual decadal prediction system" (as stated in L66). The paper that is cited does not fit here because in it, Polkova et al 2023 analyze the NA SPG skill from the WMO DCP ensemble of twelve CMIP models and would be more appropriate to be mentioned in L77.

Reply: We have changed the cited paper as suggested by the reviewer.

7. L88: Do authors have any evidence to state that anomaly initialization "became more popular"? E.g., in the WMO operational decadal prediction set, which relies on the CMIP6-based models, only 4 models (out of 12) use anomaly initializations. Anomaly initialization is an alternative and work-around method, it has been tested by some research groups, but there is no evidence that the method is getting more popular. One could compare and cite a change from CMIP5 to CMIP6 decadal predictions to support the statement that more systems became anomaly initialized. From my experience, I do not observe that they "become more popular".

Reply: We did not mean to imply that anomaly initialization was more popular than full field initialization, just that it had gained some popularity. The full paragraph has been rewritten, including the discussion about anomaly and full-field initialization, which now avoids any confusion.

8. L94-101: Are those questions actually answered? Q1: has already been answered in the Introduction. Paragraph in L101-111 suddenly reduces the scope of the study from the NA skill to that in the Labrador Sea. Authors need to introduce why they focus on the Labrador Sea all of a sudden. Q2: What method is used to answer that? Q3: Name local drivers and preconditions that will be considered in the analysis. Could question part in L94-101 be combined with the content part in L101-111? Otherwise, they duplicate each other.

Reply: We have substituted the questions by a paragraph that describes the main points that are directly addressed in the manuscript.

L110: Discussion "in light of previous studies" turned out to be very thin. L380-430: 6 summary points, 4 of them mention briefly other studies without much discussion. Moreover, the 4th section (L378) is not named as Discussion anymore.

Reply: We have expanded the discussion and in particular the contextualisation of our results with other studies. We have also modified the name of the 4th section to make it consistent with this paragraph.

9. Overall the introduction part is lengthy and dissipated. Until L90 I do not know, where it is heading. Some of the description can be squeezed substantially, for instance, it is not necessary to introduce in great detail the difference between predictions and historical simulations, it has been done in many previous papers. Paragraph L57-66 states there is no skill in the NA, while the paragraphs in L68-83 try to prove the opposite. Authors need to specify earlier in the introduction, where they are going with all this, otherwise it is not clear what of this very extensive introduction about many topics is relevant for the current study. E.g., the paragraph L41-43 could be such a place. The "main aim" (L91-92) could come earlier.

   Reply: We have restructured and streamlined the introduction, as suggested by the reviewer. Additionally, the main goal of the paper is now introduced earlier in the section to provide clearer guidance to the reader.

Methods:

1. L128: Please rephrase: Not clear what exactly the exception is with EC-Earth3. 10 members were requested and 10 members were provided; where is exception?

   Reply: In the text, this refers to the historical simulations (not predictions). For EC-Earth, these experiments were conducted by various European institutions, with experiment identifiers assigned somewhat arbitrarily. The sentence clarifies that we used the 10 historical simulations run by BSC for convenience. These do not correspond to the first 10 members (e.g., r1-10) in ESGF, which served as the criterion for the other models. The text has been rephrased to improve clarity.

2. L130: "Two models contributed with fewer than 10 members to the experiments:" Contradicts the previous sentence in L124-125. If there are two exceptions, following statement does not hold: "A total of 8 AOGCMs fulfilled all the selection criteria."

   Reply: We have revised the writing to avoid the contradiction.

3. L132: Specify difference in resolution in the text.

   Reply: Added '(0.25º and 0.4º respectively)'.

4. L136: I suggest renaming the subsection to "Verification datasets" as "three ocean reanalysis" (L138) are not "observational references" but data assimilation products - a blend of model results and observations. And adjust in the text accordingly, e.g., L150, 205, 321, etc. Also, not clear if EN4 data are actually the EN4 analysis dataset based on Optimal Interpolation. This is also not clear further on in the analysis: L137: What EN4 data are actually used, original profiles or profiles interpolated to the gridded dataset? Thus, this needs to be specified.

Reply: The subsection title was renamed to "Verification datasets" and direct mentions to observational references have been avoided. The EN4 objective analyses (optimal interpolation) were used, this has been clarified in the text.

5. L180: Is there any reason for choosing single lead years 2, 5 and 10 and not multi-year averages, which were proposed and used in previous decadal prediction studies to reduce the noise in the calculation of the prediction skill?

   Reply: We preferred to keep individual lead years for simplicity as the goal was not to maximize skill by averaging out some of the unpredictable noise but to assess how the skill changes with forecast time. We also wanted to keep a parallelism with the analyses in Costa-Carmo et al 2022, which also focused on individual forecast years.

6. Why Table 1 does not contain information about the atmospheric model and atmosphere initialization as it is probably relevant for the NAO part of the study?

   Reply: We have not included information about atmospheric initialization because it has limited relevance in decadal prediction, as the atmosphere does not contribute predictability beyond approximately two weeks. For instance, some decadal prediction systems, such as CESM, do not even initialize the atmosphere with observed states (Yeager et al., 2018). Instead, it is the ocean—and its initialization—that provides the primary source of predictability for the NAO at seasonal to multi-annual timescales.

   Reference:

   Yeager, S. G., and Coauthors, 2018: Predicting Near-Term Changes in the Earth System: A Large Ensemble of Initialized Decadal Prediction Simulations Using the Community Earth System Model. Bull. Amer. Meteor. Soc., 99, 1867–1886, https://doi.org/10.1175/BAMS-D-17-0098.1.

Results:

1. L190: "This initial skill loss might …" Some disconnect with the previous sentence, which speaks about increased skill. MPI-ESM model shows also increase of skill in the LS. Overall, this improvement seems to look very minor. To provide a quantitative estimate of improvement/loss of skill, the authors could provide percentage of grid-cells in the region that shows higher/lower skill?

   Reply: We have rephrased the text for clarity. We do not mention MPI-ESM in this context because the increase of skill with FY is only visible in the western side of the Labrador Sea box, with the eastern side showing instead a decrease in skill.

2. L191: From the figure, it does not look like "historical ensembles show comparatively higher ACC values than the DCPP at FY2". Vice versa historical simulations have a larger area of „no skill". Some models have higher correlation in the eastern NA in HIST, but again only some. To support the statement in L191, the map of the skill score or correlation difference with the significance level should be provided – that is a standard plot in similar intercomparison studies.

Reply: With that statement we referred only to models IPSL-CM6A-LR and CanESM5, the only ones for which it is actually true. We have rephrased the text to make that clearer. As suggested, we have also added the maps of the differences in correlation between the DCPP and HIST ensembles (new Figure A2), which nicely supports our statement, further illustrating how initialization is largely beneficial for the predictive skill, not only in the Labrador Sea but also in the wider North Atlantic Subpolar Gyre region.

3. L192: Suggest to change "it seems to reflect" to "it might reflect … the issue that has already been reported by …".

Reply: Done.

4. Figure 2: the color palette is a bit unfortunate: The ranges 0.1-0.3 and 0.3-0.5 are hardly distinguishable. The range 0.6-1 is not in use.

Reply: The color palette has been modified to improve visibility.

5. Figure 3: is not very informative. The arguments in paragraph (L210-215) about "cooling trends" and any "multi-annual modulations" are difficult to recognize and follow from this figure (L210-219). L210-211: Is this a trend or multi-annual variability? L217: Is the "evolution flat" or the figure is flat? Maybe a different position of the panels, which does not stretch the timeseries, could help to better transfer the message of the authors. Also, one could normalize timeseries and use the original figure in the supplementary.

Reply: We have redone the figures modifying their aspect ratio to improve the visibility of the trends. We have also rephrased the text which now avoids any general reference to multi-year variations.

6. L226: Please elaborate in the text: what does "(1) an unforced origin for the observed trend" mean?

Reply: The text has been expanded to make it more clear.

7. Figure 5: Residual correlation as in Smith et al 2019 https://doi.org/10.1038/s41612-019-0071-y might be more appropriate to separate skill due to internal variability and external forcing.

Reply: An important drawback of residual correlations is that they are only valid if the models represent the forced response correctly, and this might not hold in some regions where the HIST simulations show large disparities. We have introduced them (new Figure A3) in the context of the global maps of ACC skill, to see if they yield consistent results with new Figure A2, which displays the ACC differences between the DCPP and HIST ensembles. Interestingly, both figures highlight the Subpolar North Atlantic as the region that benefits the most from initialization. Residual correlations additionally identify the Eastern North Atlantic as another region with added value of initialization, but this result needs to be taken with caution as it is also the region with the largest ACC inter-model differences in the HIST ensemble.

8. L238: Or there are cancelling signals in the box that is analyzed, as it also includes Irminger Current and part of the North Atlantic Current in that box. From Figure 1, it follows that apart from one model, all historical experiments have skill in the Labrador Sea.

   Reply: We now mention this possibility in the text.

9. L257: What is the difference between "the local OHC skill and ultimately their forecast skill"? Please elaborate in the text.

   Reply: The text has been rephrased.

10. Figure 6: Why are there two times "depth" labels on the y-axis?

    Reply: Corrected.

11. L294, 298-303: Can one compare Figures 7 and 8? They seem to show different things: Figure 7 shows ensemble spread and Figure 8 shows shifts of action centers with lead year. Is Figure 8 plotted based on the ensemble mean or also on based on the ensemble members as Figure 7?

    Reply: They provide complementary information, so any comparison between them should be done with caution. As the reviewer correctly says, crosses in Figure 7 indicate intra-ensemble consistency in the historical runs, while in Figure 8 the circles indicate differences across forecast times in the predictions, in this case for their respective ensemble means. That is why we decided to use different symbols in each case (i.e. crosses and circles). We have amended the text and the caption of Figure 8 to make this more clear.

12. L303-306: Confusing conclusion about "centres of action appear to be unaffected by the forecast drift". Has Figure 8 been diagnosed based on the drift? But even then, Figures 7 and 8 still represent different things (ensemble spread vs. drift). Or does Figure 8 reflect temporal evolution + drift, then speaking about drift is not appropriate at all?

    Reply: We acknowledge that this sentence was imprecise. The forecast drift (as characterized by the lead time dependent climatology) has been removed prior to the computation of the EOFs that describe the centers of action. What we meant to say is that the centers of action do not seem to change with forecast time. The text has been rewritten for clarity.

13. L304: "full-field initialization does not correct the position of the simulated centers of action in models" What is the physical mechanism by which ocean initialization should correct NAO centers of action? The original hypothesis was that the NAO drives the LS variability and not the other way around. Why is atmospheric initialization not mentioned here, as these prediction systems are also initialized in the atmosphere? Also, this conclusion (L304-306) cannot be made because DCPs are not compared here with respect to the original data that have been assimilated into the respective prediction systems. Pay attention that all of these systems are

initialized from different datasets and with different assimilation/initialization methods, and not necessarily from the verification dataset that is used here. Earlier authors mentioned that ERA5 has its centers also close "to the East" (L299), CMCC-CM2-SR5 is initialized in the atmosphere with ERA-Interim and ERA5. If this is how the CMCC-model wants the centers of action to be (in HIST) and how initialization suggests (ERA-reanalyses), so why should they be located somewhere else?

Reply: We respectfully disagree with the reviewer on this point. The NAO is not solely a driver of North Atlantic Ocean variability; it also responds to changes in the North Atlantic SST patterns (e.g., Gastinneau & Frankignoul 2015). These SST patterns are expected to be corrected through full-field initialization. As already addressed in our response to point 6 for the methods section, the atmospheric initialization is less critical in this context, as the forecast times considered in our analysis are long enough for the atmospheric state to be predominantly shaped by the underlying ocean state, which has significantly more persistence. For this reason, we also maintain that our conclusion is justified even though we use a common dataset for verification (i.e., ERA5) that differs from the datasets employed for initialization in the prediction systems. We chose ERA5 because it is widely regarded as one of the most reliable and comprehensive atmospheric reanalyses, providing our best estimate of past NAO variability. Our primary goal is to evaluate which models most accurately represent the NAO centers of action and whether they change with forecast time, rather than assessing how closely the models retain their initialized atmospheric state. We believe this approach allows us to robustly identify strengths and weaknesses in how the prediction systems simulate the NAO dynamics, independent of the specific details of their initialization datasets or methods.

Reference:

Gastineau, G., and C. Frankignoul, 2015: Influence of the North Atlantic SST Variability on the Atmospheric Circulation during the Twentieth Century. J. Climate, 28, 1396–1416, https://doi.org/10.1175/JCLI-D-14-00424.1.

14. L321: Specify which observations are meant here, or is it ERA5 reanalysis?

Reply: We meant ERA5. This is not specified in the text.

15. L324: I am missing the bridge in this analysis between ocean initialization based on „full-field initialization" and „more realistic forcing of the NAO". There is no word about atmospheric initialization, more curiously, it is not even mentioned in Table 1.

Reply: Full-field initialization in the ocean helps to simulate a more realistic forcing of the NAO on the OHC by imposing a more accurate stratification, which then enables the local mixing to respond more appropriately to the atmospheric forcing associated with the NAO. We have rephrased the text to indicate this more clearly. Also, as noted in our responses to other comments, atmospheric initialization is not considered relevant in this context, as decadal predictability is determined by the more persistent oceanic state rather than the transient atmospheric state.

16. Figure 11. Caption text about the sensitivity of the stratification should be in the main text not in the caption. Instead, the range can be explained in the caption, with 0 value meaning less stratified and 1.5 - more stratified or so. In the legend the green star is not filled, what does this mean? I suggest another sensitivity test with the box confined only to the LS as e.g., in Menary et al 2016 https://doi.org/10.1002/2016GL070906, because in the very first figure almost all the models have skill in the LS. Thus, this analysis in Figure 11 contradicts Figure 1 with now half of the models suggested to not have skill in the LS. Same region, as in this study, in Hermanson et al 2014 https://doi.org/10.1002/2014GL060420 termed as western part of the SPG. Maybe authors need to reconsider naming the region as "LS" or recalculate the analysis for a smaller region that focuses only on the LS.

Reply: The part about the sensitivity to the index definition has been moved to the text and the interpretation of the index has been included in the caption, as suggested. Also, the green star in the legend has been filled, it was an oversight. We now also explicitly mention when the LS box is defined that the selected region extends into the Irminger, and that other studies have termed this whole area as the western SPG. We chose to maintain this larger area for our analysis because Figure 2 nicely illustrates that it is the entire region that exhibits significant inter-model differences, and not just its western side. Regarding the final point, we believe there is no contradiction between Figures 1 and 11, provided the region represented by the LS box is clearly defined, as it is now. Specifically, having skill in the western portion of the LS box does not imply that the whole region is predictable. For instance, the HIST ensembles for the CMCC, HadGEM, and MRI models display negative skill in the eastern side of the box, which accounts for their low ACC values in Figure 11.

17. L360: "In models that have stronger climatological surface heat fluxes in the LS". Name those models.

Reply: Done.

18. L360-363: The sentence is not clear. The ACC in the NorCPM is highly anti-correlated with the EN4, likely suggesting the opposite trend (as follows from Figure 3). What does this has to do with the " lower percentage of the observed OHC700 variance"? There are too many indirect interpretations in the last paragraphs of the result section. Please make sure to refer to figures and analysis, so that the reader can follow authors' line of thoughts.

Reply: We have rephrased the text for clarity.

19. L375, L410: In Figure 11, the multi-model mean ACC value is less than 0.5, the multi-model seems to have 0.4, which means even less variance explained than claimed in the study (25%). MRI model has the skill of 0.5.

Reply: The reviewer is right. We have corrected the numbers in the text.

Conclusions and Final Remarks

1. L383: From where does this follow that the observational uncertainties in the Labrador Sea are low?

   Reply: From Figure A1, that shows that 4 different observation/reanalysis datasets are highly correlated with each other in that region. This was also previously mentioned in lines 203-206 of the previous manuscript.

2. L390: Subpolar North Atlantic is not the region where models do not show skill in Figure 1. SPG is usually defined from 45N (or 50N) northward, while the region where the ACC has negative values in this analysis already starts at 30N (or 35N) in most of the models. It is more the region of the Gulf Stream path and the Gulf Stream separation in many models. It would be more meaningful to specify the coordinates, latitudinal bands, etc., rather than naming regions incorrectly.

   Reply: We referred to that region as the Central Subpolar North Atlantic to align with the terminology used in Carmo-Costa et al. (2022), where the negative values unequivocally occurred in that area. However, we acknowledge that in this multi-model analysis, such a definition is less precise because the location of negative ACC values varies significantly across models. We have revised the text to describe the region as being located east of the Grand Banks, which more broadly applies to all the models in this analysis.

3. L391: „It is unclear how much of this low skill is due to the large local observational uncertainties. " What is exactly meant by this sentence? That the verification dataset is uncertain and so is the skill estimate, or that the initial conditions are uncertain and thus predictions diverge too much from the true initial state? All of this might hold and not only that; model biases, initialization issues and limits of predictability are also reasons for low prediction skill. E.g., in seasonal predictions, deficiencies in predicting position of the jet stream in the atmosphere is an issue; we have the same thing in the ocean in decadal predictions, with the current's pathways (Gulf Stream separation and pathway). In this respect, it is not clear to me, why only this one reason ("local observational uncertainties") that limits skill is mentioned in the summary.

   Reply: We intended to convey that uncertainties in the observational data used for verification in this region prevent us from accurately determining the true predictive skill. However, the reviewer is correct that these observational uncertainties, which are particularly large in this region (as shown in Figure A1), most likely also affect the quality of the initial conditions, further influencing the region's predictability. We have now stated both points in the text. Regarding the other factors mentioned (e.g., model biases, initialization issues, and predictability limits), while they are valid and relevant to prediction skill in general, they are not unique to this region. For this reason, we have chosen not to include them in this specific case.

4. L399: "using multi-model approaches" for what purpose exactly? Operational climate predictions are carried out at national centers with a single model. From Figure 5, most of the models have decadal prediction skill that is higher than that of the historical simulations (L397-398). The skill score for Figure 1 could show the

quantitative difference between initialized and uninitialized simulations. As it is now, the conclusion is not convincing.

Reply: This was not a recommendation for operational centers in particular, but for climate prediction research in general. Drawing conclusions from individual models can sometimes lead to equivocal interpretations. This underscores the importance of initiatives like DCPP, which were designed to explore predictability and its drivers across multiple models, allowing for an evaluation of their consistency—or lack thereof. Our analysis of the contribution of external forcings to Labrador Sea OHC predictability exemplifies this need. Regarding Figure 5, while it is true that most models exhibit higher skill in the DCPP ensemble compared to the HIST ensemble, this difference is neither consistently significant across models nor uniform in magnitude. That is why we conclude that the added predictive value of initialization, reflected in the difference in skill between DCPP and HIST, varies across models—a conclusion we consider robust.

5.  L405: Consider this paper for discussion by Hegerl et al 2021 https://doi.org/10.3389/fclim.2021.678109 showing that models that simulate a more realistic SPG stratification show higher SST prediction skill than those that simulate less realistic stratification.

    Reply: Thanks for the reference. We now discuss it in the text.

6.  L413: Some models are initialized only in the ocean (e.g., NorCPM), others are also initialized in the atmosphere (e.g., HadGem3). Information about this is not mentioned in the manuscript. How does this difference (initializing or not initializing atmosphere, full field vs anomaly) across models affects or could affect their performance?

    Reply: As we have explained in the response to other previous comments the initialization of the atmosphere is not expected to have an important influence on the decadal predictive skill due to the very short memory of the atmosphere compared to the ocean.

7.  L420-421: From Figure 1, it does not follow that MPI-ESM model is the among best performing "in the whole North Atlantic", its skill is way less than in other models.

    Reply: We have revised the text to clarify that neither of the two high-resolution models exhibit superior performance in terms of predictive skill compared to the other models.
* * *
Minor:

1.  L29: Rephrase "help overcome"

    Reply: Done.

2.  L81: "and" not "or".

Reply: Modified as requested.

3. L211: Some problem with a sentence.

   Reply: We have rephrased the sentence.

4. L348: "Understanding uncertainties and predictability in the externally forced LS OHC700"

   Reply: We have kept our phrasing that we find to be more precise.

5. L360: "(Figure 11c)"

   Reply: Done.

6. L390: "negative skill" not "negative skill score". No "skill scores" have been shown in the manuscript.

   Reply: Done.

7. L393: "inter-model differences in terms of the skill spread". It is necessary to be precise about what differences are meant.

   Reply: Done.

---

## Author Comment (AC2)

**General comments:**

This paper analyses the variations of the ocean heat content averaged over the first 700 m (OHT700) of the ocean into an area surrounding the Labrador Sea region for the period 1970-2014. For this purpose, it uses observational data available and two multi-model sets of climate simulations, one with only the external forcing (historical simulations) and the other with decadal hindcasts starting from observation-based estimate. The analysis shows a very wide range of response in the models, especially for the historical simulations. The authors try to estimate the skill of the different systems to reproduce the OHT700 in decadal prediction and historical simulations, and found an interesting link between this skill and the capability of the models to reproduce observed mean state of stratification and ocean heat fluxes in the Labrador area.

This is an interesting and well-written paper. The analysis led is impressive given how difficult it is to deal with so many climate model data. The interpretation of the results is wise and useful, even though no definitive conclusions can be drawn from this type of multi-model analysis. At least, this is presenting an interesting intercomparison of present-day models to reproduce heat storage in the Labrador Sea area and a few interesting predictors that might be of use for observations constraints approaches. I therefore think this paper is suitable for publication. I have mainly some comments that might allow to strengthen the demonstrations and possibly improve the interpretation of the results.

Reply: We thank the reviewer, Didier Swingedouw, for his constructive comments on our manuscript. We greatly appreciate the time and effort he has invested in reviewing our work. His feedback has been invaluable in improving the quality and clarity of our paper, and we believe the revisions have significantly strengthened our manuscript.

We have carefully addressed each of the suggestions and have provided a detailed, point-by-point response to all the comments.

1. Line 35-40: here the authors are mixing discussions about the subpolar gyre and the wider North Atlantic and ocean heat content and SST. It might be worth to be a bit more specific in the description of those papers.

   Reply: We have rewritten this part of the introduction to make a more clear distinction between the regions and variables.

2. Line 61: a reference after forecast range might be useful to support this claim.

   Reply: This part of the introduction has been removed following a suggestion of the other reviewer.

3. Line 202: The LS, as represented in Figure 2, does not entirely correspond to the Labrador Sea but is going far the east, including the Irminger Sea for instance. In this respect the agreements between observation-based datasets are not that clear to the east (cf. Figure A.1), while the good agreement is taken as a reason to focus on this region in line 204. Please clarify. Have the authors tried a more tied region?

Reply: The reviewer is correct. We had not noticed that the large LS box considered includes areas where the reanalyses and EN4 show significant discrepancies in terms of OHC variability. We no longer refer to the agreement between those datasets as one of the reasons for focusing on this region. However, we do not believe that the local areas of disagreement between the observation-based datasets affect our results, as we are using regional averages and the datasets largely agree over most of the selected area. Additionally, the main reason for using this broader region, which includes both the Labrador Sea and the Irminger Sea, still holds. This area exhibits large inter-model differences in OHC skill in the DCPP ensemble across all forecast ranges, as well as in the HIST ensemble. It is also a characteristic region of deep vertical ocean mixing, with common preconditioners and drivers, whose representation varies greatly across models, contributing to the inter-model spread (as shown later in Figure 11). For these reasons, we have decided to retain the entire region in our analysis.

4. Line 249-253: ocean stratification and heat fluxes are two variables clearly linked in the convection region. If the halocline is too strong, convection is not allowed and heat fluxes can lead to sea ice formation. It might be worth to state this coupling between these two variables (maybe in the discussion).

   Reply: We now comment this in the text.

5. Line 266: it is said line 155 that density is computed with reference 1000 m (sigma_1) while in Figure 6, the caption talks about reference to the surface. Given that the numbers in Figure 6 are larger than 28, I assume this is actually sigma_1. This choice is surprising given that then authors are focusing on the very upper layer. I think it might be better to consider sigma_0 as stated in the caption (while it is not what is shown).

   Reply: The reviewer is correct: it is the caption of Figure 6 that was incorrect. Indeed, we have only used sigma_1 in the study. We understand the suggestion of using sigma_0 instead, but still believe that sigma_1 is more suitable, because we use it to understand and characterise the mixed layer in the Labrador Sea, whose influence goes beyond the upper ocean levels. We have corrected the caption in the revised manuscript.

6. Line 291-296: why are the observations not shown on Figure 7?

   Reply: We prefer to keep everything consistent within the figure, to ease its interpretation. If we included an additional panel with observations it would not have crosses as for the HIST simulations, since there is only one instance of observations as opposed to the different realizations in HIST. For that reason, we show observations elsewhere (i.e. new Figure A6).

7. Line 395-400: The use of residual ACC (Scaife & Smith 2018) might be interesting as well. I'm wondering if this might work for this type of complex quantity like OHT700, especially given the complexity of its forced response. A discussion on this aspect might be interesting here I think.

Reply: We have added a figure with the residual correlations (new Figure A3) and a discussion addressing the validity of that metric for the OHC700. To complement the residual correlations, we have also included another figure with the differences in ACC between the DCPP and HIST experiments (new Figure A2). The main takeaway from the new analysis is that residual correlations help to identify more clearly the added value of initialization in the Labrador Sea and the Eastern North Atlantic areas, which is less evident when directly comparing the DCPP and HIST ACC maps. We also note that the results of the residual ACC need to be interpreted with caution, as some areas show large inter-model uncertainties in terms of OHC skill for the HIST ensemble. This implies that some models do not correctly represent the observed forced signal, a requirement for the residuals correlation to be meaningful.

8. Line 412-416: I have the feeling that this aspect has not been much depicted in the result section, so that this discussion seems a bit coming out of the blue. Maybe useful to add a few points on this in the results section.

Reply: We now address the differences in OHC skill between full-field and anomaly-initialized prediction systems in the discussion around Figure 1. Also, the specific benefits of each initialization approach for the representation of mean stratification in the Labrador Sea and the local NAO forcing at different forecast times are discussed in their respective results sections.

9. Line 431: Yes, the omission of advective processes is clearly missing in this paper, but I can understand that it is far from easy to have those quantities from such a large ensemble of simulations. What about citing Ortega et al. (2015) that was also discussing this type of processes in details? + typo at "mechanisms".

Reply: Thanks for spotting the typo and for suggesting the reference. The typo has been corrected, and the reference is now cited.

**References**

Scaife A.A., Smith D., 2018. A signal-to-noise paradox in climate science. npj Clim. Atmos. Sci., 1, 28. doi: 10.1038/s41612-018-0038-4

Ortega P., et al. (2015) Reconciling two alternative mechanisms behind bidecadal AMOC variability. Progress in Oceanography 137, pp. 237-249.

---

## Author Response (AR2)

**Response Reviewer #2**

The authors have accounted for my comments and have done a very good job to produce an almost publishable manuscript.

Reply: We thank the reviewer for the positive feedback and the additional comments.

I only have a few minor comments that the authors might find of interest.

**1. Line 35: this last statement has not been proved properly in Caesar et al. (2021). To be able to attribute a signal to anthropogenic forcing, you need to compare it with forced signal from model simulations. This has been done in Latif et al. (2022) and Bonnet et al. (2021) who both concluded that internal variability might be more likely to explain the long term trend found in Caesar et al. (2018), not the external forcing. This might be useful to state this. Observed Greenland melting is unlikely to also explain this (e.g. Devilliers et al. 2021). The next paragraph is exploring the role of internal variability, but I think it is misleading since the focus will be mainly on interannual variability, up to decadal variations, while the statement from the paragraph before clearly discusses multi-decadal trend, which might be driven by different processes. I think it might be useful to clarify the different time scale involved and the relevance of this work in this respect.**

Reply: We thank the reviewer for the feedback. We now cite all the articles suggested, to give a more complete discussion of the recent AMOC weakening. We also clarify in the following paragraph that the discussion refers to interannual to decadal internal variability.

**2. Line 324 : "sea-ice" should be replaced by "sea ice" as used in the rest of the manuscript when it is referring to a noun (e.g. line 345)**

Reply: Done

**3. Line 354: As stated later on, the forced signal from HIST cannot be compared per se with observations, as those ones include both signals (forced and internal). Maybe it's worth clarifying this already here. Also, the motivation of this last part is not very clear to me in the general narrative of the paper. I wonder if a better transition with the former part, or a slight introduction to the motivation of this one might not be useful.**

Reply: These are two very good points. We have included a sentence to motivate this final section and also expanded the paragraph to clarify why the forced signal from HIST cannot be directly evaluated from observations.

**4. Line 358 and caption of Fig. 11: It might be worth discussing the FT here. We can see on Fig. 5 that this is almost constant for HIST. Is it entirely or are there some edge effects (maybe not since the time of analysis has been chosen to avoid this). Please clarify. Also the use of ACC for HIST is surprising and might need to be discussed since it is not that straightforward to understand (we usually look into trends when analysing HIST and observations).**

Reply: We confirm that the ACC values for the HIST experiments remain constant across all forecast years because the evaluation period used for their computation is fixed. This information is now explicitly indicated in the figure caption. We now also mention the long-term trends in the discussion of Figure 11. We also clarify that we chose to evaluate the forced signals in the models using the ACC metric, as the response to external forcings is not accurately captured by a simple linear trend, particularly due to the episodic nature of several volcanic eruptions during the evaluation period. However, we acknowledge that the main results shown in Figure 11 would likely hold if the metric were replaced by the linear trend, as Figure 4 demonstrates a strong relationship between trend-based skill and ACC.

**5. Line 416: this last point might be better substantiated if observations were also indicated as a grey overlap for instance in Fig. 11, to highlight that models with best ACC in OHC in HIST have very poor representation of stratification and heat fluxes which is actually surprising results when compared to Hegerl et al. (2021)**

Reply: Observations are indicated as a dashed black line on the Figure, and support the message that we make in that line.

**6. Line 421: "(Quin et al., 2020)" should be "Quin et al. (2020)"**

Reply: Corrected

**7. Line 431: I have the feeling that Fig. 5 does not show this, with the skill of NorCPM1 being not significant from FY 4. Can you please clarify or correct this statement?**

Reply: The reviewer is right. Figure 5 clearly shows that this statement does not apply to the whole Labrador Sea region. We have rephrased the text, which now specifically mentions the westernmost side of the Labrador Sea, for which the added value of initialization persists up to forecast year 10 (Figure 1).

**Response Reviewer #3**

The authors have addressed the reviewers' suggestions thoroughly, and the manuscript is in good shape for acceptance. However, I recommend a minor revision before final acceptance to improve clarity and accuracy.

Reply: We thank the reviewer for the positive feedback and the additional comments.

Specific comments:

**1. For the benefit of future readers who may not be well-acquainted with the decadal prediction system, I suggest adding a few sentences explaining how the DCPP simulations are performed, as well as the methodology behind the Historical simulations in CMIP6. This information is not explicitly stated in the manuscript and would help improve understanding for future readers.**

Reply: This is a fair point. We have reorganised and extended the paragraph in the introduction where both DCPP and HIST simulations are explained.

**2. Line 180: The parentheses in "(figure A3)" is missing and should be corrected.**

Reply: Corrected

**3. Line 203: The reference to the "western Subpolar Gyre" seems inappropriate, given that the described region, including the Irminger Sea, is significantly larger than what is typically considered the western Subpolar Gyre. If this designation is based on a specific reference, further justification is needed. Otherwise, I recommend deleting this statement and avoiding the term "western Subpolar Gyre" throughout the manuscript. The use of "LS region" is acceptable.**

Reply: We have removed the sentence

**4. Figure 3 Caption: There is a typo in the caption where "for" appears incorrectly. Please revise accordingly.**

Reply: Corrected